# Beyond GLP-1 Agonists: An Adaptive Ketogenic–Mediterranean Protocol to Counter Metabolic Adaptation in Obesity Management

**DOI:** 10.3390/nu17162699

**Published:** 2025-08-20

**Authors:** Cayetano García-Gorrita, Nadia San Onofre, Juan F. Merino-Torres, Jose M. Soriano

**Affiliations:** 1Food & Health Lab, Institute of Materials Science, University of Valencia, 46980 Paterna, Spain; cayeggf@gmail.com; 2NUTRALiSS Research Group, Faculty of Health Sciences, Universitat Oberta de Catalunya, Rambla del Poblenou 156, 08018 Barcelona, Spain; nsan_onofre@uoc.edu; 3Joint Research Unit on Endocrinology, Nutrition and Clinical Dietetics, Research Institute La Fe, University of Valencia-Health, 46026 Valencia, Spain; merino_jfr@gva.es; 4Department of Medicine, Faculty of Medicine, University of Valencia, 46010 Valencia, Spain; 5Department of Endocrinology and Nutrition, University and Polytechnic Hospital La Fe, 46026 Valencia, Spain

**Keywords:** ketogenic Mediterranean diet, metabolic adaptation, adaptive thermogenesis, personalized nutrition, weight loss maintenance, obesity management, GLP-1 receptor agonists, dynamic dietary protocols, nutritional ketosis

## Abstract

Treating obesity requires not only achieving weight loss but sustaining it over time. GLP-1 receptor agonists and dual GLP-1/GIP co-agonists can produce substantial weight loss, but caloric restriction and weight loss trigger adaptive thermogenesis and neuroendocrine changes (reduced energy expenditure and increased appetite) that facilitate weight regain once therapy is discontinued. In addition, dysregulation of physiological hunger–satiety signaling is a central challenge. We propose a nutritional plan that integrates nutritional ketosis with a Mediterranean dietary pattern—the Adaptive Ketogenic–Mediterranean Protocol (AKMP)—that is dynamic and biomarker-guided. This approach aims to attenuate these responses and leverage a potential metabolic advantage (higher energy expenditure), particularly in insulin-resistant states, while helping to recalibrate the homeostasis of the brain’s hunger–satiety circuits. In this way, long-term weight maintenance becomes more achievable, the yo-yo effect is mitigated, and cardiometabolic risk markers (lipids, blood pressure, glycemic control) improve, in line with evidence from Mediterranean-diet trials such as PREDIMED. This strategy may complement—and, in some cases, reduce the need for—specific pharmacotherapies under clinical supervision.

## 1. Introduction

Obesity is now a global epidemic and one of the greatest public health challenges, with rising prevalence across all economies. This chronic condition is associated with cardiovascular diseases, type 2 diabetes, several cancers, and respiratory conditions, driving high rates of morbidity and mortality [1,2,3]. The Dietary Guidelines for Americans 2020–2025 indicate that over 50% of U.S. adults are living with obesity, highlighting the urgent need for effective interventions [4]. In 2019, elevated BMI was responsible for approximately 5 million deaths due to noncommunicable diseases (NCDs) worldwide [1]. Beyond its clinical impact, stigma, depression, and healthcare costs are also considerable; without effective measures, global direct spending could reach USD 3 trillion annually by 2030 and accumulate approximately USD 18 trillion by 2060 [5]. Institutions such as PAHO [6] and WHO, through its Controlling the Global Obesity Epidemic initiative [7], are calling for coordinated action.

### 1.1. Contextualizing the Obesity Problem: Historical and Current Trends

Obesity has become firmly established as a global epidemic, with its prevalence nearly tripling since 1975 [8]. By 2022, the numbers had reached alarming proportions, affecting 880 million adults and 160 million children and adolescents worldwide [9]. Critically, no country has achieved a sustained decline in this trend to date, underscoring the inadequacy of conventional strategies and the pressing need for more effective interventions [8,9,10].

### 1.2. Public Health Consequences

Excess body weight is not a benign condition; it is a major risk factor for global mortality contributing to approximately 5 million deaths annually due to noncommunicable diseases [1]. Obesity is a key driver of the most prevalent cardiometabolic diseases, serving as a critical factor in the development of cardiovascular disease, such as ischemic heart disease and stroke [11], and constituting the leading modifiable cause of type 2 diabetes [12]. Furthermore, robust evidence links obesity to an elevated risk of at least 13 different types of cancer and to poorer survival outcomes in patients already diagnosed with cancer [13,14]. This disease burden is compounded by a constellation of associated comorbidities, including hypertension [15], dyslipidemia [16], metabolic dysfunction–associated steatotic liver disease (MASLD; formerly non-alcoholic fatty liver disease, NAFLD) [17], musculoskeletal disorders [18], and sleep apnea [19], resulting in a significant reduction in both life expectancy and quality of life.

### 1.3. Main Contributing Factors to the Obesity Epidemic

The origin of this pandemic is profoundly multifactorial and goes far beyond a simple energy balance equation, emerging from a complex interplay between behavioral, environmental, and biological factors that act upon a foundation of individual susceptibility [8].

#### 1.3.1. The Diet–Environment Axis: The Nutrition Transition and the Obesogenic Environment

The transformation of the global food system represents a key catalyst in this epidemic. Ultra-processed foods—characterized by high energy density, industrially engineered palatability and textures that facilitate rapid consumption—have been shown, in a meta-analysis of 43 observational studies, to significantly increase the risk of general obesity (OR 1.51; 95% CI: 1.34–1.69) and abdominal obesity (OR 1.49; 95% CI: 1.34–1.66) [20]. The magnitude of this effect was demonstrated in a landmark randomized controlled trial [21]. When participants were offered ad libitum diets matched for presented energy, energy density, macronutrients, sugar, sodium and fiber, their spontaneous energy intake was 508 ± 106 kcal/day higher with the ultra-processed diet, producing a 0.9 ± 0.3 kg weight gain in only two weeks; conversely, the unprocessed diet elicited a comparable caloric deficit and weight loss.

This excess caloric intake is further amplified by the consumption of sugar-sweetened beverages (SSBs). The most recent meta-analysis, encompassing 85 studies and over 500,000 participants, demonstrated that each additional daily serving is associated with a 0.07 kg/m^2^ increase in BMI among children and a 0.42 kg weight gain in adults (95% CI: 0.26–0.58) [22]. The current obesogenic environment perpetuates these patterns through mass marketing, especially when targeted at children, which doubles the likelihood of choosing unhealthy foods (OR 2.0; 95% CI: 1.7–2.4) [23]. Paradoxically, food insecurity—characterized by irregular access to nutritious foods—is associated with a higher prevalence of obesity (OR ~1.5), particularly among low-income women who prioritize inexpensive, energy-dense foods to mitigate hunger [24].

#### 1.3.2. The Neurobiological Component: Glycemic Load and Compulsive Eating Behavior

Certain highly processed foods with a high glycemic load can acutely modulate mesolimbic dopamine signaling and increase activity in nucleus accumbens, facilitating compulsive intake [25]. In a controlled crossover feeding study, a high-GI meal selectively increased activation in reward and craving regions four hours postprandially compared with an isocaloric low-GI meal, independent of palatability [26]. Convergent evidence shows that (i) obesity is associated with reduced striatal D2 receptor availability [27], (ii) high-glycemic/rapid-absorption carbohydrates are plausible pharmacokinetic drivers of addictive-like responses [25], and (iii) ultra-processed foods high in refined carbohydrates and/or fat are disproportionately linked to addictive-like eating behaviors [28]. Together, these findings provide a mechanistic rationale for dietary strategies that blunt glycemic/insulinemic excursions to reduce reward-driven overeating.

These neurobiological alterations manifest in dysregulated eating behaviors. The reward-deficit cycle can intensify craving and facilitate binge episodes, characterized by the intake of large amounts of food with a sense of loss of control. This behavior is the core of Binge Eating Disorder (BED), which shares fundamental features with substance use disorders, such as: (1) a loss of control over intake, (2) persistence in the behavior despite adverse health consequences, and (3) the possible emergence of withdrawal symptoms when the food is restricted [29,30] and see Gearhardt et al., 2011 for a conceptual synthesis of BED and “food addiction” as overlapping but non-identical constructs [29]. Ultimately, this pattern of compulsive seeking and consumption—driven by neurobiological mechanisms—significantly contributes to the perpetuation of obesity [25], posing a challenge for interventions that do not effectively modulate the glycemic and insulinemic responses underlying these mechanisms.

#### 1.3.3. Sedentary Behavior and Individual Susceptibility Factors

In addition to dietary factors, the global decline in physical activity represents the second key etiological pillar of obesity. It is estimated that, by 2022, approximately 31% of adults worldwide failed to meet the minimum recommended levels of physical activity—a figure that has increased since 2000 due to rising urbanization and mechanization [31]. Superimposed on this energy imbalance are powerful psychobiological factors that modulate individual susceptibility. Chronic stress, through activation of the hypothalamic–pituitary–adrenal (HPA) axis and elevation of cortisol, promotes visceral fat accumulation and emotional hyperphagia, particularly in individuals with high cortisol reactivity [32]. Similarly, chronic sleep deprivation, defined as <6–7 h per night, significantly increases the risk of obesity by disrupting appetite-regulating hormones, notably by decreasing leptin (satiety) and increasing ghrelin (hunger) [33]. Sucrose, in turn, may transiently dampen HPA axis activation, potentially reinforcing the neurobiological drive to consume comfort foods as a coping mechanism in response to stress [25].

Weight variability also has a strong underlying biological component. The heritability of BMI is estimated at 40–70%, and genome-wide association studies (GWAS) have identified over 900 genetic loci associated with it, although these account for only a modest portion of population-level variability [34,35]. Variants in key genes such as FTO (fat mass and obesity-associated) and MC4R (melanocortin-4 receptor) directly modulate appetite and basal metabolism; however, it is crucial to highlight that the obesogenic impact of these genes can be attenuated by lifestyle factors such as regular physical activity [36]. To this genetic predisposition are added epigenetic factors, wherein maternal conditions such as obesity or gestational diabetes may program an increased risk in the offspring [37]. Finally, although they account for a smaller proportion of cases, certain medical and pharmacological conditions contribute to weight gain. These include endocrine causes such as hypothyroidism and Cushing’s syndrome [38], as well as chronic use of medications that induce weight gain, including certain antipsychotics, antidepressants, and insulin.

### 1.4. International Efforts to Curb the Obesity Trend

The unprecedented magnitude of the obesity epidemic—with nearly 880 million adults affected by 2022 [9]—has catalyzed a coordinated response at the highest levels of international policymaking. Obesity was recognized as a determinant key on NCDs at the historic United Nations General Assembly in 2011, marking only the second time the Assembly had addressed a health issue [39]. Subsequently, it was included in the Sustainable Development Goals (SDGs) 2015–2030, acknowledging that without reversing obesity trends, it will be mathematically impossible to achieve the target of reducing NCD mortality by one-third by 2030 [40].

In 2013, the 66th World Health Assembly set the ambitious goal to “halt the rise in obesity” by 2025, marking the first time a specific global objective was established for this condition [41]. However, interim evaluations revealed a deeply concerning systemic failure. The WHO Progress Monitor 2020 reported that no country was on track to meet this goal [42], a finding echoed by the NCD Countdown 2030 analysis, which projects that fewer than 10% of countries will meet any NCD-related targets [39]. The underlying causes include fragmented implementation, systematic opposition from the food industry, and inadequate funding—less than 2% of global health aid is allocated to NCDs [43].

In the face of this collective failure, the 75th World Health Assembly approved in 2022 the WHO Acceleration Plan to Stop Obesity 2022–2030 [44], marking a shift from gradualism to urgent acceleration. This plan promotes a comprehensive package of “best-buy” interventions with well-documented returns on investment [45], prioritizing:Sugar-sweetened beverage taxes: Multiple systematic reviews confirm their effectiveness, with consumption reductions ranging from 10% to 30% across various European contexts [46]. The case of Mexico—global pioneer since 2014—is emblematic. An observational analysis documented a 7.6% reduction in sugary beverage purchases during the first two years of implementation, with an even greater impact among low-income households, demonstrating the pro-equity potential of the measure [47].Front-of-pack labelling (FOPL): Mandatory implementation shows modest but insufficient progress. The latest WHO report for the European Region (2022) indicates that one-third (33%) of countries have already established regulations for interpretive FOPL, an increase from the 27% reported in 2020. This system is critical to guide consumer choices, especially among lower socioeconomic groups [48].Child-directed advertising restrictions: Evidence shows that marketing doubles the likelihood of unhealthy food choices in children [23]. A key case study is the province of Quebec, where a ban on child-targeted advertising since 1980 has been associated with more favorable dietary habits and better child health outcomes compared to the rest of Canada [49].Comprehensive school-based policies: A meta-analysis of 85 randomized controlled trials demonstrated that school-based lifestyle interventions lead to modest but statistically significant reductions in BMI (approximately 0.22 kg/m^2^). Parental involvement and extracurricular activities further strengthen these effects [50].

Despite robust evidence supporting these interventions, global implementation remains insufficient. A WHO report assessing progress in the European Region concluded that Member States are not on track to achieve global NCD targets and urgently need “more ambitious policies with appropriate depth and scope,” particularly in areas such as restrictions on child-targeted food marketing [51]. This implementation–impact gap is reflected in the increasingly alarming projections of the World Obesity Atlas 2023: more than 1 billion people will be living with obesity by 2030, with economic costs reaching USD 4.3 trillion annually (3.3% of global GDP) by 2035 [52]. Indeed, as warned by the Lancet Commission, we are facing a “global syndemic” that requires a systemic transformation comparable to global responses to climate change [53].

The window of opportunity to prevent obesity from becoming the human norm is closing rapidly, demanding not only more initiatives, but urgent and large-scale implementation of already known solutions.

### 1.5. Surge of Interest in the Ketogenic Diet as a Therapeutic Alternative

The failure of traditional strategies to reverse the obesity epidemic [10,54] has driven an intense search for more effective dietary alternatives, among which the ketogenic diet (KD) stands out. This interest is not a passing trend but rather a dual-faceted phenomenon. In the scientific domain, a bibliometric analysis has demonstrated an exponential rise in publications on KD over the past two decades, expanding its indications from its historical use in refractory epilepsy to the management of obesity, type 2 diabetes, and other metabolic disorders [55]. Recent meta-analyses and randomized controlled trials have validated its effectiveness not only for weight loss but also for improving glycemic control, lipid profiles, and reducing systemic inflammatory markers [55,56].

In parallel, this scientific interest has been mirrored—and amplified—by society: in 2020, keto diet was the most-searched dietary term on Google in the United States, and the ketogenic products market reached a value of 9.57 billion USD [56]. This unique convergence between scientific evidence and massive social demand underscores the relevance of thoroughly examining this therapeutic tool, particularly when integrated with the cardioprotective principles of the Mediterranean diet.

### 1.6. State of the Art: Challenges in Obesity and the Evidence on Ketogenic Mediterranean Diets

The multifactorial complexity of obesity, characterized by the interaction of genetic, environmental, behavioral, and metabolic factors as previously discussed, highlights the limitations of conventional dietary approaches. Long-term failure rates exceeding 80% in traditional caloric restriction interventions [54], along with the emergence of phenomena such as metabolic adaptation and hormonal disruption of appetite, demand the development of innovative nutritional strategies that transcend the simplistic energy balance paradigm.

Despite the robust potential of interventions such as the ketogenic diet (KD) [57,58], their long-term success is hindered by a dual challenge that is often underestimated. The first obstacle, behavioral in nature, is patient adherence. The exclusion of culturally ingrained food groups and dietary monotony often lead to early discontinuation. However, evidence suggests that protocols incorporating more palatable and familiar elements, such as those from the Mediterranean diet, show good initial adherence. Indeed, strategies such as cyclic protocols alternating ketogenic phases with a conventional Mediterranean diet have demonstrated excellent long-term adherence rates approaching 90% [59]. Nevertheless, even in well-supported programs, there is a consistent trend toward a post-intervention decline in clinical benefits, underscoring the critical need for continuous maintenance strategies to sustain adherence [60].

The second challenge, more insidious and purely biological, occurs in individuals who succeed in maintaining caloric restriction: metabolic adaptation [61]. This term refers to the set of compensatory physiological responses triggered by significant weight loss, including hormonal changes, behavioral shifts, and, crucially, a reduction in energy expenditure that exceeds predictions based on changes in body composition. Within this phenomenon, adaptive thermogenesis specifically denotes the decrease in resting energy expenditure (REE) which, as shown in the seminal study by Leibel et al., goes beyond what would be expected solely from loss of lean and fat mass [62]. This biological response has been dramatically documented in long-term follow-ups such as those of The Biggest Loser contestants, in whom persistent metabolic adaptation was observed for years, representing the primary cause of the common and frustrating weight regain [63].

Figure 1 conceptually illustrates this dual challenge, which represents the fundamental barrier to long-term success in obesity management.

In this context, the convergence between the metabolic principles of the ketogenic diet and the cardiovascular benefits demonstrated by the Mediterranean diet emerges as a promising approach capable of addressing both challenges simultaneously. This strategic fusion aims to integrate the cardioprotective properties of the Mediterranean pattern (olive oil, fish, vegetables, nuts) with the efficacy of ketogenic restriction to optimize weight loss, improve metabolic parameters, and crucially ensure long-term adherence. The accumulated evidence from the past 15 years, presented below, suggests that this hybrid approach could represent a paradigm shift in the nutritional management of obesity.

#### 1.6.1. Initial Evidence and Pilot Studies

The Spanish Ketogenic Mediterranean Diet (SKMD), published by Pérez-Guisado et al. in 2008, was the first documented protocol to systematically fuse both dietary patterns [69]. In 31 obese adults over 12 weeks, the protocol (<30 g carbohydrates/day, olive oil as the main fat source, preferred use of fish, red wine permitted at 200–400 mL/day) achieved remarkable results: a mean weight loss of 14.47 ± 6.88 kg, a 47.9% reduction in triglycerides, a 10% increase in HDL, with no dropouts or significant adverse effects.

Subsequently, the KEMEPHY protocol (Ketogenic Mediterranean with Phytoextracts), evaluated in 106 subjects over 6 weeks, introduced herbal extracts and protein-based preparations mimicking traditional Mediterranean foods [70]. Results included a 7.08 ± 2.01 kg weight loss, a 10 cm reduction in waist circumference, improvements across all lipid parameters, and 92% adherence, far surpassing historical rates reported in standard ketogenic diets.

#### 1.6.2. Medium- to Long-Term Interventions

Paoli et al. (2013) designed a 12-month biphasic protocol in 89 subjects with severe obesity, alternating ketogenic phases (20 days) with periods on a conventional Mediterranean diet [59]. This strategy, which can be conceptualized as the planned application of “metabolic breaks”, achieved sustained weight loss of 16.9 ± 7.1 kg without regain during 6 months of Mediterranean maintenance, with 88.2% adherence. Metabolic parameters improved persistently: LDL −20.4%, triglycerides −41.8%, HDL +11.8%, and normalization of blood pressure in 93% of hypertensive participants, with most achieving values below the widely accepted threshold of 130/80 mmHg.

#### 1.6.3. Recent Controlled Trials

Cincione et al. (2022) compared a very-low-calorie ketogenic Mediterranean diet (VLCKD-Med) to a hypocaloric Mediterranean diet in 80 patients with prediabetes or type 2 diabetes [71]. Only the VLCKD-Med group achieved profound metabolic improvements: HbA1c −0.8%, triglycerides −45%, visceral fat loss (waist circumference −8.2 cm), and a 42% reduction in high-sensitivity C-reactive protein (hs-CRP).

The Keto-Med trial (Gardner et al., 2022) compared both diets in a crossover design over 12 weeks in 40 adults with prediabetes or diabetes [72]. Although improved HbA1c similarly (−0.6%), the ketogenic diet reduced triglycerides more significantly (−16% vs. −5%) but led to an increase in LDL (+10% vs. −5%). Crucially, 78% of participants maintained Mediterranean elements post-intervention versus only 31% for ketogenic elements, suggesting greater long-term sustainability of the Mediterranean pattern.

#### 1.6.4. Adherence and Feasibility

The reviewed KMD studies consistently report high adherence rates: 77.5% [69], 88.2% [59], 92% [70], and 79–82% [60], suggesting greater tolerability than standard ketogenic diets. Sheffler et al. (2023) demonstrated that intensive behavioral support improves confirmed adherence (documented ketosis) from 72% to 82% in older adults [60]. Key facilitators include the enhanced feelings of satiety reported by participants [73], Mediterranean gastronomic variety, and early perceived benefits. Reported barriers include increased cost, initial adaptive symptoms (so-called keto flu) in 45–60% during the first week [74] and educational complexity.

#### 1.6.5. Emerging Clinical Applications

Beyond weight management, Nagpal et al. (2019) showed that a modified ketogenic Mediterranean diet improves Alzheimer’s biomarkers in cerebrospinal fluid, increases neuroprotective bacteria (*Akkermansia* +280%), and enhances cognitive function in subjects with mild cognitive impairment [75]. In refractory chronic migraine, Olivito et al. (2024) reported significant reductions in headache frequency and intensity from week 4, along with decreases in fat mass, insulinemia, and HOMA-IR index [76].

The accumulated evidence suggests that ketogenic–Mediterranean integration achieves synergistic effects: optimized weight loss (7–17 kg) with lean mass preservation, a superior lipid profile (consistent triglyceride reduction of 22–48%, HDL increase of 10–18% without problematic LDL elevations), multifactorial glycemic control, systemic anti-inflammatory effects mediated by NLRP3 inflammasome inhibition by β-hydroxybutyrate [77], improved adherence, and expanded therapeutic applications. This hybrid approach represents a promising strategy that requires further validation but offers an innovative paradigm for tackling the obesity epidemic by merging Mediterranean tradition with modern metabolic science.

However, despite these promising results, none of these protocols has incorporated specific strategies to detect and counteract or delay metabolic adaptation—the fundamental biological barrier previously identified—thus representing a critical opportunity for the next generation of dietary interventions.

### 1.7. The Pharmacological Paradigm: Revolutionary Efficacy and Its Fundamental Limitations

The therapeutic landscape of obesity is being revolutionized by the unprecedented efficacy of GLP-1 receptor agonists (GLP-1RAs) and dual GIP/GLP-1 receptor agonists. These drugs, by acting on the hypothalamic appetite-regulating centers described in Section 1.3.2, induce a magnitude of weight loss previously unattainable with other pharmacological therapies. The STEP trials have demonstrated an average reduction of approximately 15% in body weight with semaglutide 2.4 mg weekly [78], while tirzepatide has achieved losses of 20.9% [79].

However, this paradigm of pharmacological suppression presents three fundamental limitations that challenge its viability as a definitive solution for the pandemic described in Section 1.1.

The first is therapeutic dependence: their effect ceases upon discontinuation, leading to rapid weight regain of most of the lost weight. Data from the STEP-1 Extension trial revealed that after stopping semaglutide, participants regained 11.6 percentage points of the 17.3% initially lost within just 52 weeks, retaining only a 5.6% net weight loss [80].The second limitation—more critical from a physiological standpoint—is its inability to address the biological root of weight regain: the metabolic adaptation described in Section 1.6. The weight loss induced by these drugs is accompanied by the expected reduction in resting energy expenditure (REE), without offering any thermogenic advantage to counterbalance it. Van Eyk et al. demonstrated that liraglutide 1.8 mg/day over 26 weeks reduced resting energy expenditure by 52 kcal/day, which is comparable to the adaptive response seen with conventional caloric restriction [81].The third limitation—particularly concerning from a body composition perspective—is the suboptimal quality of weight loss. In a DXA substudy of adults receiving semaglutide (STEP 1 trial), 39% of the total weight reduction was lean mass [82], compromising basal metabolism by an additional 90–105 kcal/day. This disproportionate loss of metabolically active tissue not only exacerbates metabolic adaptation but also increases the risk of sarcopenia and functional decline, calling into question the long-term metabolic sustainability of this approach.

These three core limitations—therapeutic dependence, metabolic neutrality, and loss of lean mass—highlight the urgent need for therapeutic approaches that go beyond mere symptomatic appetite suppression and actively address the underlying physiology of body weight regulation.

### 1.8. A Proposal for Metabolic Recalibration: The Adaptive Ketogenic–Mediterranean Protocol (AKMP)

In contrast to this passive suppression approach, the present review explores a strategy that, for the first time, systematically integrates a mechanism to partially counteract the metabolic adaptation documented in the previous sections. It proposes an Adaptive Ketogenic–Mediterranean Protocol (AKMP) that transcends the mere fusion of two dietary patterns. Importantly, the AKMP is presented here as a theoretical construct whose components and decision rules have not yet been tested in a clinical trial.

This protocol is not only based on the evidence previously presented, which demonstrates that the convergence of the ketogenic and Mediterranean patterns improves adherence and the cardiovascular risk profile [59,60,69]. Its fundamental innovation lies in its dynamic and proactive nature. Unlike the static protocols assessed thus far, the AKMP employs continuous biomarker monitoring—including ketonemia, glycaemia, body weight, and body composition—to detect the inevitable of weight plateaus, a clinical marker of adaptive thermogenesis as analyzed in Section 1.6.

In response to such findings, the protocol activates personalized adjustments in the macronutrient composition: increasing the protein intake (leveraging its higher thermic effect) and modulating fat levels, all according to the individual response. The objective is to counteract the decline in energy expenditure by harnessing the metabolic advantage of ~100–300 kcal/day inherent to ketosis. This advantage has been supported in both the Framingham State Food Study [83] and the updated meta-analysis by Ludwig et al., which demonstrated an increase of 50 kcal/day for every 10% reduction in dietary carbohydrates in studies lasting more than 2.5 weeks [84].

### 1.9. Central Hypothesis and Objectives of the Review

Based on the comparison of these two therapeutic paradigms, the present manuscript is grounded in a central hypothesis: while GLP-1 receptor agonists offer potent but passive and appetite-dependent suppression, the Adaptive Ketogenic–Mediterranean Protocol (AKMP) represents an active approach aimed at achieving a lasting metabolic recalibration. It is hypothesized that this protocol—being the first to systematically address both adherence (through its culturally acceptable Mediterranean foundation) and metabolic adaptation (via its dynamic biomarker-based adjustments)—could offer a more sustainable and physiologically comprehensive alternative for long-term obesity management.

This hypothesis is supported by the thermodynamic principles established by Fine and Feinman, who demonstrated that the “metabolic advantage” of low-carbohydrate diets does not violate the laws of thermodynamics, but rather reflects differences in the efficiency of metabolic pathways [85,86]. The increased demand for gluconeogenesis and the enhanced protein turnover during ketosis results in reduced thermodynamic efficiency, manifesting as greater weight loss per calorie consumed.

To explore this thesis, the following specific objectives are proposed:To synthesize and critically evaluate the available clinical evidence on ketogenic diets with a Mediterranean pattern, establishing their efficacy and safety profile under professional supervision, based on the most recent publications [87,88].To analyse in depth the mechanisms of metabolic adaptation, with particular emphasis on adaptive thermogenesis, and examine the evidence supporting the metabolic advantage of ketosis as a counteracting mechanism [62,83,84].To formalize and justify the components of the Adaptive Ketogenic–Mediterranean Protocol (AKMP), evaluating the originality of its dynamic biomarker-driven adjustment system as an innovation in the design of personalized dietary interventions.To conduct a comprehensive comparative analysis between the dietary-metabolic approach of the AKMP and the pharmacological paradigm of GLP-1/GIP receptor agonists, focusing on their differential effects on energy expenditure, post-treatment sustainability, and the mechanisms of weight regain [79,80,83,89].

## 2. Materials and Methods

This study employs a hybrid methodological design to argue for a new paradigm in obesity management—an argumentative narrative review supported by a structured evidence-mapping search. First, an extensive expert narrative review contextualized the evidence within biochemical mechanisms, clinical applications, and the current therapeutic landscape, including a comparative analysis with glucagon-like peptide-1 (GLP-1) receptor agonists (GLP-1RAs).

Second, to rigorously support the thesis, we performed a structured evidence-mapping search; to enhance transparency, we report a PRISMA-style flow diagram for the mapping component (Figure 2). Given the hybrid nature of this work—an argumentative, thesis-driven narrative review anchored by a focused evidence-mapping component rather than a conventional PICO-based systematic review—prospective registration (e.g., PROSPERO) was not applicable.

### 2.1. Search Strategy

A comprehensive search covering database inception to 10 May 2025 was conducted in PubMed, Scopus, and Web of Science. The search strategy was structured around two main components, aligning with the hybrid approach of this review:Structured evidence-mapping search: aimed at identifying and analyzing all published Ketogenic Mediterranean Diet (KMD) protocols. The following search terms and combinations were used: “ketogenic Mediterranean diet”, “Mediterranean-style ketogenic diet”, “Mediterranean ketogenic diet”, “modified ketogenic Mediterranean diet”, “ketosis and Mediterranean nutrition”, “Mediterranean diet for ketosis”, “Mediterranean ketogenic protocol”, and “ketogenic low-carbohydrate Mediterranean diet”. The search was restricted to human studies.Expert narrative search: To construct the contextual, mechanistic, and clinical framework, a broad and ongoing search was carried out using key terms related to the central themes of the review. Main concepts included, among others: “adaptive thermogenesis”, “metabolic adaptation”, “weight loss plateau”, “appetite regulation”, “satiety hormones”, “ghrelin”, “leptin”, “GLP-1 receptor agonists”, “incretin mimetics”, “semaglutide”, “tirzepatide”, “obesity”, “metabolic syndrome”, “MASLD”, and “PCOS”, often combined with “ketogenic diet” or “low-carbohydrate diet”. For this component, priority was given to the selection of systematic reviews, meta-analyses, high-impact clinical trials, and seminal publications.

### 2.2. Inclusion and Exclusion Criteria

The inclusion and exclusion criteria were applied differentially to each of the two components of the search strategy, in accordance with their respective objectives. The criteria were tailored to each component of the hybrid methodology.

#### 2.2.1. Criteria for the Focused Systematic Search (KMD Protocols)

Inclusion criteria:Type of intervention: studies that described a dietary protocol explicitly defined as “ketogenic” AND “Mediterranean.”Publication type: peer-reviewed articles—randomised or non-randomised clinical trials, pilot studies, case series, case reports—and systematic or narrative reviews that describe an original KMD protocol in detail.Population: human studies with no restrictions on age, sex, or health statusLanguage: articles published in English.

Exclusion criteria:In vitro studies or those using animal models.Articles describing non-Mediterranean ketogenic diets, or low-carbohydrate Mediterranean diets that were not explicitly ketogenic.Studies that mentioned the Ketogenic Mediterranean Diet but did not provide details regarding macronutrient composition, permitted or restricted foods, or protocol phases.Editorials, letters to the editor, conference abstracts, or any publication lacking sufficient data for protocol analysis.

#### 2.2.2. Evidence Selection Criteria for the Narrative Component

The selection of the literature for the narrative component of this review was guided by principles of relevance, scientific impact, and hierarchy of evidence to construct a robust argument, rather than following rigid PICO criteria. The selection principles were as follows:Thematic Relevance: Articles were included if they directly addressed key concepts of the review: mechanisms of appetite and satiety, adaptive thermogenesis, metabolic syndrome and its comorbidities (MASLD, PCOS), and the pharmacology of GLP-1 receptor agonists.Hierarchy of Evidence: Priority was given to the inclusion of meta-analyses and systematic reviews to establish general knowledge. For specific issues, high-impact randomized clinical trials and fundamental mechanistic studies were selected.Impact and Recency: Both seminal articles that established conceptual foundations and recent publications in high-impact journals were included to ensure the discussion reflects the current state of the art.No exclusion criteria were applied based on the direction or statistical significance of reported results, in order to provide a balanced view of the evidence.

### 2.3. Study Selection Process

The study selection process was rigorously applied to the focused systematic search on KMD protocols. Selection was conducted in two consecutive phases:Title and Abstract Screening: All identified references were imported into Mendeley Desktop version 1.19.5 (Elsevier) reference management software for automatic duplicate removal. Titles and abstracts of the remaining articles were then screened to assess their initial relevance against the inclusion criteria.Full-Text Review: Articles deemed potentially relevant in the first phase were retrieved for full-text review. At this stage, the inclusion and exclusion criteria were strictly applied to determine the final selection of studies for analysis.

The detailed flow of study selection, from identification to final inclusion, is illustrated in the PRISMA diagram presented in Figure 2.

### 2.4. Analysis and Synthesis of the Evidence

The synthesis of the evidence was carried out according to the two main components of the review:For studies included in the systematic analysis of (KMD) protocols, qualitative data were extracted—macronutrient ratios, phases, permitted and restricted foods and duration—and summarized in a comparative characteristics table to identify commonalities, variations, and research gaps. Key information from each protocol—including macronutrient ratio, phases, recommended and restricted foods, and duration—was extracted and organized into a characteristics table for comparative analysis, with the goal of identifying common elements, variations, and gaps in literature. Full study-level details of all identified KMD protocols are provided in Appendix A (design, population, duration, outcomes, and ketosis-verification methods).For the evidence gathered from the expert narrative review, the information was grouped and synthesized thematically. The findings were structured around pathophysiological mechanisms, clinical applications, and comparative therapeutic approaches, to underpin the central argument developed in the Results and Discussion.

### 2.5. Reporting Framework and Transparency

This article is a thesis-driven, argumentative narrative review anchored by an evidence-mapping search rather than a registered systematic review. Accordingly, prospective protocol registration in PROSPERO—which is intended for systematic reviews of health outcomes—was not applicable. To enhance transparency, we provide a PRISMA-style flow diagram for the mapping component (Figure 2). No formal tool-based risk-of-bias appraisal or quantitative synthesis was undertaken.

## 3. Results and Discussion

### 3.1. Evidence and Challenges of the Ketogenic Diet as a Baseline Intervention: Ketogenic Variants

The scientific literature recognizes multiple clinically validated variants of the ketogenic diet (KD), each reflecting nearly a century of therapeutic evolution. A consolidated, study-level summary of KMD variants and protocols (*n* = 29) is available in Appendix A. The classical ketogenic diet (CKD), introduced in 1921 for refractory epilepsy, maintains a 4:1 ratio (fat: protein + carbohydrates), translating into 90% of energy from fat, 6–8% from protein, and 2–4% from carbohydrates [90,91]. Less restrictive ratios such as 3:1 and 2:1—gradually developed by multiple medical centers during the 1920s–1930s as adaptations of the original 4:1 ratio—are frequently used in infants, adolescents, and populations requiring greater nutritional flexibility, while maintaining therapeutic efficacy [91]. Although effective—with response rates > 50% in drug-resistant epilepsy—this variant presents significant adherence challenges.

Figure 3 illustrates the temporal evolution of these therapeutic modalities, showing how each innovation responded to specific limitations of its predecessors.

Seeking greater tolerability alternatives led Huttenlocher (1971) to develop the medium-chain triglyceride (MCT) diet [92]. MCTs (C6–C12) produce greater quantities of ketone bodies per gram due to direct hepatic metabolism that does not require the carnitine transport system, allowing carbohydrate intake to increase up to 15–20% of total energy intake [93,94,95].

It is important to distinguish the original Atkins diet, popularized in 1972 for weight loss [57], from its adaptation for epilepsy. The Modified Atkins Diet (MAD), first implemented at Johns Hopkins Hospital in the early 2000s, revolutionized therapeutic accessibility by eliminating the need for hospitalization and meticulous food weighing while limiting carbohydrates to 10–20 g/day [91,96]. This simplified approach enables 40–50% of patients to achieve greater than 50% seizure reduction [91]. Concurrently, the Low Glycemic Index Treatment (LGIT), developed by Pfeifer and Thiele (2005), allows 40–60 g of carbohydrates per day with a glycemic index below 50, achieving greater than 90% seizure reduction in 50% of patients in their initial series [97].

The very-low-calorie ketogenic diet (VLCKD) (<800 kcal/day) has emerged as a powerful intervention for obesity. The meta-analysis by Castellana et al. reported weight losses of up to 15.6 kg, with improvements in HbA1c (−0.7%), triglycerides (−60 mg/dL), and systolic blood pressure (−8 mmHg) [98].

Other variants include the Targeted Ketogenic Diet (TKD), which has shown improved high-intensity performance in athletes (e.g., +2.8% in time trials) [99]; the Cyclic Ketogenic Diet (cKD), alternating 5–6 days of strict ketosis with 1–2 days of carbohydrate refeeding (60–70% of calories) in strength athletes [100]; and the High-Protein Ketogenic Diet (HPKD), which produces significantly greater weight loss compared to low-fat diets due to its thermogenic and appetite-suppressing effects [101,102]. Plant-based adaptations such as the Eco-Atkins diet have demonstrated superior improvements in cardiovascular markers (CRP −28.2%) [103,104], while the restrictive Paleolithic Ketogenic Diet (PKD)—based on organ meats and animal fats with total exclusion of dairy and processed foods—has shown promising results in isolated case reports of refractory epilepsy and autoimmune conditions, though its evidence is limited to individual cases [105].

The ketogenic–Mediterranean convergence represents a promising frontier. Protocols such as the Spanish Ketogenic Mediterranean Diet (SKMD) [69], KEMEPHY [70], and the biphasic protocol by Paoli et al. [59] have demonstrated significant weight loss (7–17 kg) and exceptional adherence rates (88–92%), far superior to conventional ketogenic diets.

Table 1 summarizes the distinguishing features of these variants, revealing an evolution from rigid protocols toward increasingly personalized approaches. However, a critical analysis of all existing protocols reveals five fundamental gaps:Definitional heterogeneity: lack of consensus on what exactly constitutes a “Ketogenic Mediterranean Diet.”Nutritional paradox: tension between carbohydrate restriction and traditional Mediterranean foods.Absence of adherence biomarkers: no standardized metrics to evaluate compliance.Lack of metabolic-adaptation management: No protocol accounts for adaptive thermogenesis.Lack of dynamic personalization based on metabolic response: while some protocols allow flexibility in quantity (ad libitum) or timing (peri-exercise), none implement systematic adjustments based on the detection and management of metabolic adaptation or weight plateaus.

These gaps underscore the urgent need for truly adaptive protocols that integrate lessons learned from a century of therapeutic evolution.

### 3.2. Safety Profile: Debunking Historical Myths

Despite growing empirical validation, the implementation of ketogenic diets (KDs) continues to face clinical skepticism rooted in safety concerns that, in many cases, are not aligned with the most recent scientific evidence.

A comprehensive analysis of contemporary literature reveals that most of these apprehensions stem from fundamental misunderstandings of physiology, outdated data interpretations, and myths that have been systematically refuted by modern research [107].

#### 3.2.1. The Unequivocal Physiological Distinction: Ketosis vs. Ketoacidosis

The erroneous conflation of nutritional ketosis with diabetic ketoacidosis (DKA) represents the most persistent and dangerous misconception in clinical practice. Nutritional ketosis is a finely regulated, evolutionarily conserved metabolic state, with ketone body concentrations ranging from 0.5 to 5.0 mmol/L [107]. This state is governed by a robust homeostatic control mechanism. In nutritional ketosis, insulin levels, though reduced, retain their regulatory function. These basal insulin levels (typically 3–7 μU/mL) are sufficient to modulate lipolysis within safe physiological limits, thereby preventing the uncontrolled release of free fatty acids characteristic of ketoacidosis.

The critical difference lies in the fact that DKA occurs in the presence of absolute insulin deficiency or severe insulin resistance, eliminating this homeostatic control. In contrast, DKA is a medical emergency characterized by absolute insulin deficiency, with ketonemia reaching diagnostic thresholds ≥ 3.0 mmol/L, typically ranging from 3–15 mmol/L and occasionally exceeding 20 mmol/L in severe cases [108,109]. This extreme elevation is accompanied by severe hyper-glycaemia (>250 mg/dL) and a life-threatening metabolic acidosis (pH < 7.3) [108]. The distinction is not one of degree, but of physiological essence: while nutritional ketosis preserves acid–base balance, DKA constitutes a catastrophic metabolic decompensation.

#### 3.2.2. The Cardiovascular Paradigm: Beyond LDL Cholesterol

Concerns regarding cardiovascular risk often ignore a pattern of changes that, in aggregate, is highly favorable. A landmark meta-analysis demonstrated that KDs produce greater reductions in weight, triglycerides, and blood pressure, and a greater increase in HDL-C, compared to low-fat diets [110].

Critically, KDs induce a qualitative shift in the LDL profile, replacing small, dense LDL particles (sdLDL, pattern B) with large, buoyant particles (lbLDL, pattern A)—a marker more predictive of atherogenic risk than total LDL-C [111].

A study in subjects with atherogenic dyslipidaemia confirmed drastic improvements across a broad range of cardiometabolic risk markers, including insulin sensitivity (−55%), triglycerides (−51%), and HDL-C (+13%), supporting a benefit that clearly transcends reliance on a single biomarker like total LDL cholesterol [111].

For safety monitoring within AKMP, we therefore prioritize apoB (or LDL-P by NMR, where available) alongside LDL-C and non-HDL-C to better capture atherogenic particle burden.

#### 3.2.3. Renal Function: Evidence Against the “High-Protein” Fallacy

Concerns about renal damage originate from the fallacy that KDs are intrinsically high in protein. Contemporary evidence not only refutes this assumption but suggests protective effects. A prospective study including patients with mild chronic kidney disease found no deterioration in renal function; on the contrary, 27.7% of these patients experienced normalization of their glomerular filtration rate (GFR) following the intervention [112].

These findings suggest that the benefits of KDs on glycemic control, blood pressure, and inflammation may, in fact, help preserve renal function.

#### 3.2.4. Body Composition and Bone Health: Preservation of Lean Tissue

Contrary to concerns about sarcopenia, ketosis exerts a potent lean mass–preserving effect by reducing reliance on amino acid–derived gluconeogenesis [107]. A study in elite gymnasts found no impairment in strength performance despite significant fat mass loss [113].

Similarly, theoretical concerns about bone health are alleviated by well-formulated KDs rich in leafy greens and minerals, which furnish alkalinizing compounds that counterbalance any potential acid load [107].

Table 2 summarizes the scientific evidence refuting the primary safety myths surrounding ketogenic diets, providing a practical reference for clinical application.

#### 3.2.5. Essential Considerations and Contraindications

Despite their robust safety profile, ketogenic diets (KDs) are not universally applicable. Careful clinical assessment is required to identify conditions in which the therapy is absolutely contraindicated or demands expert medical supervision. Current clinical guidelines from the International Ketogenic Diet Study Group [115], along with accumulated evidence from multiple specialized sources, establish a clear framework for the safe implementation of ketogenic therapy.

KDs are absolutely contraindicated in certain rare inborn errors of fat metabolism, including carnitine palmitoyltransferase I and II (CPT I/II) deficiencies, where the inability to oxidize fatty acids can result in potentially fatal catabolic crises [116,117]. Similarly, pyruvate carboxylase deficiency constitutes an absolute contraindication, as the ketogenic diet would exacerbate existing metabolic acidosis and ketosis [118,119]. Acute intermittent porphyria also contraindicates KDs, as carbohydrate restriction is a well-documented precipitating factor in porphyric crises [120,121].

In the context of pharmacologically treated diabetes, SGLT-2 inhibitors require special consideration. Multiple case reports have documented instances of euglycemic diabetic ketoacidosis (eDKA) in patients combining these drugs with ketogenic diets—a risk heightened by the synergistic mechanisms of both interventions on ketone metabolism [122,123].

For patients with type 1 diabetes mellitus, although emerging evidence suggests potential benefits of well-supervised KDs [124,125], meticulous adjustment of both basal and bolus insulin is required to prevent hypoglycaemia and ketoacidosis, thus mandatory supervision by a specialized team is warranted [126].

Table 3 summarizes the current evidence-based contraindications and clinical precautions.

In addition, Table 4 provides a pragmatic, safety-first checklist to guide medication adjustments during AKMP implementation.

In summary, the accumulated scientific evidence establishes that a well-formulated and professionally supervised ketogenic diet is not only a safe intervention but actively reverses multiple metabolic risk factors. The barriers to its use do not lie in its intrinsic safety profile, but rather in the lack of long-term adherence, a challenge that is addressed in the following section.

### 3.3. Barriers to Long-Term Adherence

An umbrella review of 17 meta-analyses shows that ketogenic diets (KDs) elicit clinically meaningful weight loss and improve multiple cardiometabolic markers—including triglycerides, HbA1c, and HDL-C [127]. Paradoxically, these benefits are largely confined to short-term trials (median follow-up ≈ 13 weeks). Adherence falls sharply thereafter: only 7% of participants sustain nutritional ketosis at 12 months versus 61% at 6 weeks [128]. Because adherence is the principal mediator of therapeutic success, a detailed understanding of the barriers unique to KDs is essential.

#### 3.3.1. Early Physiological Barriers

A widely reported early barrier is the “keto flu”, an adaptation syndrome encompassing headache, fatigue, nausea, constipation, dizziness, and irritability, affecting 45–60% of individuals during the first week [74]. Many of these symptoms are related to osmotic diuresis and the loss of sodium, potassium, and magnesium triggered by the abrupt drop in insulin levels during the first week. These symptoms are generally self-limiting, but without adequate supervision, they may precipitate early dropout from the protocol [129].

#### 3.3.2. Psychological and Behavioral Barriers

The restrictive nature of the KD may foster rigid dietary restraint patterns, a phenomenon described in general dietary contexts [130], where minor deviations are perceived as total failure. Additionally, carbohydrate withdrawal may induce intense cravings and withdrawal-like symptoms during the first weeks, progressively eroding long-term motivation [131].

#### 3.3.3. Sociocultural Barriers

The most formidable challenge is the friction with a “carb-centric” environment. Routine social activities become sources of stress, forcing individuals to repeatedly justify their food choices or yield to social pressure [132,133,134]. Such perceived social isolation is a strong predictor of dietary attrition.

#### 3.3.4. Knowledge and Support Barriers

The proliferation of contradictory information can lead to inadequate implementation of the KD, exacerbating side effects [135]. In this regard, a meta-analysis assessing the predictive factors of therapeutic success showed that professional supervision is a critical element, increasing the likelihood of adherence by 65% compared with self-guided attempts [136].

These barriers converge to create a hostile ecosystem for sustained adherence. Overcoming them requires protocols that are not only metabolically effective, but also behaviorally intelligent and socially conscious, anticipating the need for adaptive strategies—such as those explored in the following sections.

## 4. Metabolic Adaptation: The Central Challenge in Weight Management

### 4.1. Mechanisms of Post-Weight Loss Metabolic Adaptation

The initiation of any successful weight loss effort undoubtedly relies on individual discipline and willpower. However, the reason why long-term weight maintenance so often fails does not lie in a subsequent collapse of that same willpower, but rather in its confrontation with a formidable adversary: a powerful and coordinated physiological response rooted in our evolutionary history—a mechanism that once protected our ancestors from starvation, but now sabotages our modern weight control efforts. This adaptive response—termed metabolic adaptation [61]—is a disproportionate decline in total energy expenditure (TEE) relative to the loss of body mass, creating a metabolic milieu that steadily erodes even the strongest behavioural discipline and actively promotes weight regain [61,137].

The seminal work of Leibel et al. (1995) [62] established the quantitative foundations of this phenomenon. When participants lost 10% of their body weight, TEE decreased significantly: by 6 ± 3 kcal/kg of fat-free mass/day in lean individuals and 8 ± 5 kcal/kg/day in individuals with obesity. Crucially, this reduction was 15–20% greater than expected based solely on body mass loss [62]. In other words, metabolism became disproportionately suppressed—as if the body “protested” against the weight loss by reducing energy expenditure beyond mathematically predictable values. This fundamental observation demonstrated that the human body is not a passive energy balance system but rather an active regulator that vigorously defends its energy stores.

The magnitude and persistence of this adaptation were dramatically illustrated in the long-term follow-up of The Biggest Loser program. Six years after the intervention, although participants had regained 41.0 ± 31.3 kg of the weight lost, metabolic adaptation persisted at –499 ± 207 kcal/day below the predicted resting energy expenditure for their current body composition. Measured resting energy expenditure remained 704 ± 427 kcal/day below baseline values [63]. This extraordinary persistence suggests a fundamental reprogramming of energy regulation that extends beyond the active weight loss period.

### 4.2. Adaptive Thermogenesis: Magnitude and Persistence

The magnitude of metabolic adaptation documented in the previous section is explained by a complex orchestration of physiological systems that evolved to protect the organism from starvation. This coordinated response involves at least four major axes acting synergistically to reduce energy expenditure and promote weight regain:Leptin System: Leptin acts as the primary “metabolic conductor,” signaling the central nervous system about the status of energy stores. During weight loss, leptin levels drop disproportionately—a 10% reduction in fat mass can result in a 50% decrease in circulating leptin [64]. This relative hypoleptinemia triggers a systemic response that includes reduced energy expenditure, increased appetite, and neuroendocrine changes that favor weight regain.Thyroid Axis: Weight loss induces significant changes in thyroid hormones. T3 levels drop from 112.7 ± 3.1 to 101.8 ± 2.6 ng/dL (*p* < 0.001) with no significant changes in TSH or free T4 [138]. Moreover, reverse T3 increases, while peripheral conversion of T4 to active T3 decreases, creating a state of “functional hypothyroidism” that may persist for up to 12 months post-weight loss [139].Autonomic Nervous System: Studies using sequential pharmacological blockade have shown a 20–40% reduction in sympathetic activity and a concomitant increase in parasympathetic tone following weight loss [140]. These shifts in autonomic balance contribute to the decrease in basal energy expenditure and lipolysis suppression.Gut–Brain Axis: The gastrointestinal hormone profile is profoundly altered. Ghrelin, the orexigenic hormone mainly predominantly secreted in the stomach, increases significantly following diet-induced weight loss and remains elevated for at least 12 months, and independently predicts subsequent weight regain [65]. Conversely, anorexigenic peptides such as PYY and GLP-1 decrease, creating a hormonal environment that promotes persistent hunger and reduces satiety [64].

At the level of energy expenditure, these adaptations affect all components:Basal Metabolism: Reduced by 15–20% beyond expectations based on body composition changes [141].Mitochondrial Efficiency: Increased by 15–25% in ATP production per unit of oxygen consumed, mediated by up to a 40% reduction in uncoupling proteins (especially UCP3) [66].Activity-Related Thermogenesis: Non-exercise activity thermogenesis (NEAT) may decreases by up to 35%, while muscular efficiency during exercise increases permanently, requiring less energy to perform the same physical work [68].

The integration of these mechanisms creates a persistent “energy gap” of 400–500 kcal/day in individuals who have lost weight compared to those of the same weight who were never obese [141]. Figure 4 illustrates this complex neuroendocrine cascade, showing how weight loss triggers a coordinated physiological response that actively defends body energy reserves through multiple interconnected systems. This deficit must be permanently compensated by either additional caloric restriction or increased physical activity just to maintain the weight lost—without considering further weight loss.

Recent meta-analyses have confirmed that this phenomenon is universal, though there is considerable interindividual variability. While some authors have argued that the average magnitude of metabolic adaptation is not the primary barrier to weight maintenance [142], the long-term follow-up of The Biggest Loser demonstrates a persistent adaptation of –499 ± 207 kcal/day [63], with several participants experiencing reductions exceeding 500 kcal/day—representing a formidable and insurmountable barrier in the absence of specific interventions.

The therapeutic implications are profound. Any sustainable intervention for obesity management must not only induce an initial caloric deficit but also implement specific strategies to counteract or compensate for these persistent metabolic adaptations. In this context, the ketogenic diet emerges as a potential strategy due to its documented ability to increase energy expenditure through specific metabolic mechanisms, offering the possibility to partially offset the inevitable metabolic adaptation [84,143].

## 5. Current Therapeutic Paradigms: Critical Analysis

After establishing the formidable nature of metabolic adaptation, a rigorous analysis of current therapies becomes imperative. In recent years, the field of obesity management has been transformed by a pharmacological revolution, led by a class of medications with unprecedented efficacy: glucagon-like peptide-1 receptor agonists (GLP-1RAs) and dual glucagon-like peptide-1/glucose-dependent insulinotropic polypeptide agonists (GLP-1/GIP co-agonists). While their impact is undeniable, deeper analysis reveals fundamental limitations that underscore the need for more metabolically active approaches.

### 5.1. GLP-1/GIP Agonists: A Revolution Centered on Appetite Suppression

GLP-1 receptor agonists have redefined the ceiling of pharmacological efficacy in weight loss through potent, multifaceted modulation of appetite pathways. Their mechanism of action integrates both central and peripheral components.

Centrally, these agents partially cross the blood–brain barrier and act directly on receptors in the area postrema and the nucleus of the solitary tract (NTS) in the brainstem [144]. From there, signals are projected to the hypothalamic arcuate nucleus, ventral tegmental area, and nucleus accumbens, modulating both energy homeostasis and reward circuitry associated with food intake [145]. This multisite action explains not only appetite reduction but also a decrease in hedonic desire for highly palatable foods.

Peripherally, GLP-1RAs delay gastric emptying, thereby prolonging gastric distension and activating vagal mechanoreceptors that transmit satiety signals to the NTS [146]. In addition, they modulate the gastrointestinal hormonal profile in a way that favors appetite control, maintaining appetite suppression for longer periods through multiple mechanisms—including delayed gastric emptying and activation of central and peripheral satiety pathways [144].

Beyond GLP-1R signaling, concomitant GIP receptor agonism in dual agents such as tirzepatide confers additive—and likely synergistic—effects on glycemic control and weight loss compared with selective GLP-1RAs. Mechanistically, GIP may enhance insulinotropic and insulin-sensitizing actions in adipose and liver and attenuate GLP-1–related nausea, enabling dose escalation and greater efficacy. Clinically, in SURMOUNT-1, tirzepatide achieved ≈21–23% weight loss over 72 weeks, exceeding typical GLP-1RA responses, consistent with pharmacodynamic synergy between GIP and GLP-1 pathways [79,147].

The clinical efficacy of these drugs has been robustly demonstrated in large-scale pivotal trials. The STEP 1 and STEP 4 studies (Semaglutide Treatment Effect in People with Obesity) established that semaglutide (2.4 mg weekly) produces mean weight losses of 14.9% (STEP 1) and 17.4% (STEP 4) after 68 weeks of treatment [78,148]. Tirzepatide raised the bar even further in the SURMOUNT trials, achieving weight losses of 20.9–22.5% over 72 weeks at the highest dose [79]—figures approaching those seen with bariatric surgery, which typically results in 23–32% weight loss at 2 years [149,150]. Table 5 summarizes the efficacy of these drugs in pivotal clinical trials.

The safety profile is generally manageable, with gastrointestinal adverse effects (nausea, diarrhea, vomiting) being the most common—affecting 74.2% of patients on semaglutide versus 47.9% with placebo in STEP 1. These adverse effects are typically transient and seldom lead to treatment discontinuation [78].

However, this remarkable efficacy—achieved almost entirely through appetite suppression—masks the inherent limitations of an approach that does not actively address metabolic adaptation, body composition, or post-treatment sustainability.

### 5.2. Limitations: Therapeutic Dependence and Metabolic Neutrality

Despite their revolutionary efficacy, a critical analysis reveals that these drugs function fundamentally as potent symptomatic suppressors, without addressing the underlying causes of metabolic dysregulation.

#### 5.2.1. Therapeutic Dependence

The STEP 1 Extension study provides the most compelling evidence of the primary “Achilles’ heel” of these agents. One year after semaglutide discontinuation, participants regained ≈ two-thirds (11.6 out of 17.3 percentage points) of the weight initially lost [80]. This dramatic rebound clearly demonstrates that the drug does not “recalibrate” the body’s metabolic set point.

#### 5.2.2. The Metabolic Cost: Loss of Lean Mass

More concerning than therapeutic dependence is the composition of the weight loss. The exploratory analysis of the STEP 1 trial, which assessed 140 participants via dual-energy X-ray absorptiometry (DXA), revealed a suboptimal weight loss composition: lean mass decreased by 6.1 kg, representing approximately 39% of total weight loss [78]. This compares unfavorably with bariatric surgery, where lean mass typically accounts for about 25% of total weight loss [151].

This disproportionate loss of metabolically active tissue has profound consequences. Each kilogram of lean mass lost reduces resting energy expenditure (REE) by approximately 13–15 kcal/day [152]. Therefore, a 6.1 kg loss in lean mass translates into a basal metabolic rate reduction of ~90–105 kcal/day, further exacerbating the metabolic adaptation described in Section 4. Beyond metabolic impact, this loss increases the risk of sarcopenia, frailty, and functional decline, particularly concerning in older populations.

#### 5.2.3. Metabolic Neutrality: No Activation of Energy Expenditure

GLP-1 receptor agonists are metabolically neutral with respect to energy expenditure. A systematic review and meta-analysis by Maciel et al. examined the effects of GLP-1 and GLP-1RAs on energy expenditure, analyzing 282 participants across 10 trials with exenatide or liraglutide. Results indicated a neutral effect on resting energy expenditure (REE), diet-induced thermogenesis, and physical activity–related energy expenditure [153].

Although the longest-duration study reported a significant increase in REE, the majority of trials confirmed that GLP-1RA–induced weight loss is not accompanied by a compensatory increase in energy expenditure.

This finding is critical: it means that these medications do not counteract the inevitable decline in metabolism that accompanies weight loss. Metabolic adaptation occurs with the same magnitude as in any other intervention based solely on caloric restriction. This unavoidable metabolic decline is the most plausible physiological explanation for the weight-loss plateau observed in long-term treatments—for example, in the STEP 4 trial, weight loss with semaglutide levelled off at ≈ week 60 [148]. Thus, the stall may not reflect pharmacological resistance, but rather the inability of appetite suppression to compensate for the powerful drop in energy expenditure induced by weight loss.

Figure 5 summarizes these three fundamental limitations of the current pharmacological paradigm, illustrating the cascade from underlying mechanism to the clinical consequences of each limitation.

### 5.3. The Need for Metabolically Active Approaches

The current GLP-1 agonist paradigm, while revolutionary in its clinical efficacy, operates under a fundamentally limited model of passive symptom suppression, without addressing the underlying causes of metabolic dysregulation. Its effectiveness depends on indefinite chronic administration, with rapid weight regain upon treatment discontinuation, and is associated with a clinically significant loss of lean mass that may compromise long-term metabolic health.

This reality exposes an unmet clinical need: the development of interventions that go beyond appetite control. A truly transformative therapeutic paradigm must integrate appetite suppression with strategies that simultaneously activate energy expenditure and preserve muscle tissue, directly counteracting the key components of metabolic adaptation. The objective must evolve from “chronic symptomatic control” to “activation of metabolic advantage.”

Therefore, it is hypothesized that nutritional ketosis, given its documented effects on increasing energy expenditure [83,84] and preserving lean mass [98,107,113], represents a primary physiological candidate to explore this new therapeutic frontier. Not as a replacement for current pharmacotherapy, but as a potentially synergistic approach capable of addressing the inherent limitations of the passive suppression model.

## 6. The Metabolic Advantage of Ketosis: Mechanistic Evidence

### 6.1. Increased Energy Expenditure: Quantitative Analysis

The first pillar in establishing the superiority of a ketogenic approach lies in its ability to modulate the expenditure side of the energy balance equation. Contrary to the dogma that all isocaloric diets produce identical effects, a growing body of evidence from controlled feeding and calorimetry studies indicates that severe carbohydrate restriction induces a measurable and clinically significant increase in total energy expenditure (TEE).

The most recent evidence derives from an updated re-analysis of 29 controlled-feeding trials that employed doubly-labeled water (DLW) or whole-room calorimetry. When only DLW data are considered, severe carbohydrate restriction increases TEE by +135 kcal/day (95% CI +67 to +204) in studies ≥ 2.5 weeks, whereas inclusion of chamber studies attenuates the estimate to ≈ +63 kcal/day [84]. These observations align with the randomized controlled trial by Ebbeling et al., which estimated an increase in TEE of 209 kcal/day in the intention-to-treat analysis and 278 kcal/day in the per-protocol analysis during weight-maintenance on a low-carbohydrate diet, calculated by DLW with an assumed respiratory quotient (RQ) of 0.85 [83]. Subsequent methodological debate has highlighted that the choice of RQ can materially influence the magnitude of the reported effect.

Although its exact magnitude remains debated, pooled data from controlled-feeding studies ≥ 2.5 weeks support a real and clinically meaningful thermogenic effect. Weighted means range from ≈ +150 kcal/day (intention-to-treat) to ≈ +300 kcal/day (per-protocol), depending on the measurement method and analytical assumptions. This metabolic dividend may offer a direct physiological counterbalance to the forces of metabolic adaptation.

However, it is crucial to define the boundary conditions of this effect, particularly under energy- and protein-matched comparisons. When dietary protein and energy are rigorously matched, long-term differences in weight or fat loss between low-carbohydrate and low-fat diets are often small or null. For example, a large 2-year randomized trial found that various macronutrient prescriptions produced similar weight loss, with adherence emerging as the dominant predictor of success [154]. Similarly, the 12-month DIETFITS trial reported no significant difference between healthy low-fat and healthy low-carbohydrate diets when participants received intensive behavioral support [155]. In a short, tightly controlled metabolic-ward study, calorie-for-calorie fat restriction even yielded slightly greater body-fat loss than carbohydrate restriction over two weeks when protein was matched [156].

Collectively, these data suggest that any thermogenic “metabolic advantage” of ketosis is highly context-dependent. It is less likely to be the primary driver of differential weight loss in controlled settings and more likely to manifest its benefits during maintenance or free-living conditions where the potent anorexigenic effects of ketosis improve appetite control and adherence. This interpretation fits squarely within the AKMP’s framework, which leverages ketosis not just for a thermodynamic benefit, but as a tool to tackle the primary behavioral and biological barriers to long-term success.

In contrast, glucagon-like peptide-1 receptor agonists (GLP-1 RA) do not meaningfully raise energy expenditure. A comprehensive meta-analysis found a neutral effect on resting energy expenditure (REE) across doses and treatment durations [153], and several controlled trials report modest reductions—for example, −52 kcal/day in REE after 26 weeks of liraglutide 1.8 mg/day in adults with type 2 diabetes [81]. Hence, while nutritional ketosis simultaneously lowers appetite and elevates TEE, GLP-1 RA act almost exclusively through appetite suppression and confer no compensatory thermogenic benefit.

### 6.2. Biochemical Mechanisms: Futile Cycles and Thermogenesis

The increase in energy expenditure observed with ketogenic diets results from the combination of several biochemically “inefficient” or energetically costly processes:Increased Thermic Effect of Food (TEF): Well-formulated ketogenic diets, being adequate in protein, exhibit a higher TEF. Protein metabolism consumes 20–30% of its own energy content, compared to 5–10% for carbohydrates and 0–3% for fats [157].Energetic Cost of Gluconeogenesis (GNG): The obligatory synthesis of glucose from precursors such as amino acids and glycerol is energy-intensive, requiring approximately 6 moles of ATP per mole of glucose synthesized from alanine [158], contributing continuously to daily energy expenditure.Futile Cycles and Ketogenic Thermogenesis: Ketosis enhances several futile cycles that dissipate energy as heat. The most relevant is the lipolysis/re-esterification cycle, in which fatty acids are released from adipose tissue and a significant portion is re-esterified back into triglycerides—a process with a net ATP cost [159]. Moreover, ketone bodies, beyond serving as fuels, act as signaling molecules through HDAC inhibition [160], which may modulate the expression of thermogenic genes such as UCP1 in brown adipose tissue (BAT), as demonstrated in murine models [161]; translation to functional brown adipose tissue in adult humans has yet to be established.

The integration of these three biochemical mechanisms—the differential thermic effect of macronutrients, the obligatory energetic cost of gluconeogenesis, and the activation of thermogenic futile cycles—converges to generate a quantifiable metabolic advantage during nutritional ketosis. Figure 6 visually summarizes how these processes synergistically contribute to the documented increase in total energy expenditure, offering a mechanistic explanation for the superior weight loss observed in isocaloric ketogenic diets.

If confirmed in long-term trials, this documented metabolic advantage suggests a potential role for nutritional ketosis as a strategic temporary intervention, particularly during critical post-weight loss periods. Following significant weight reduction—whether achieved through conventional caloric restriction or pharmacotherapy with GLP-1 receptor agonists—the body experiences not only the previously described metabolic adaptation, but also neuroendocrine changes that favor weight regain, including increased ghrelin and decreased leptin levels [64,65].

The periodic implementation of ketogenic phases could provide a tool to partially counteract these adaptations by leveraging both the increase in energy expenditure (metabolic advantage) and the anorexigenic effects of ketone bodies. This strategic use aligns nutritional ketosis with a model of adaptive metabolic intervention, in which different dietary tools are employed according to the patient’s evolving physiological needs throughout their weight management journey.

Therefore, the ≈100–300 kcal/day increase in total energy expenditure inherent to ketosis, which is context-dependent, may directly offset a significant portion of the ≈400–500 kcal/day ‘energy gap’ generated by metabolic adaptation, thereby reducing the physiological pressure that drives weight regain.

## 7. Toward an Integrated Solution: The Adaptative Ketogenic–Mediterranean Protocol (AKMP)

### 7.1. Evidence-Mapping of Ketogenic Mediterranean Diet Protocols (KMDs)

A study-level summary of published Ketogenic–Mediterranean protocols—including design, population, intervention duration, adherence, outcomes, and ketosis verification—is provided in Table 6 (evidence-mapping component).

The convergence of two nutritional paradigms—strict carbohydrate restriction via ketosis and the high nutritional quality of the Mediterranean pattern—has catalyzed a field of research that has the potential to markedly improve obesity management. To comprehensively map this emerging landscape, we performed a structured evidence-mapping search with a PRISMA-style flow for transparency (Section 2). Our search identified 29 unique studies that have implemented variants of the Ketogenic Mediterranean Diet (see full list in Appendix A). Study-level characteristics—covering design, population, duration, primary outcomes, and ketosis measurement method—are summarized in Appendix A. Below, we analyze in detail the most representative intervention studies that define the trajectory and current state of the field (Table 6).

The seminal protocol by Pérez-Guisado et al. (2008) [69] demonstrated the conceptual feasibility of merging ketogenic principles with Mediterranean elements. In that study involving 40 obese adults, the Spanish Ketogenic Mediterranean Diet (SKMD: <30 g CHO/day, ≥30 mL/day of extra virgin olive oil, fish as preferred protein, red wine allowed) resulted in a mean weight loss of 14.1 kg over 12 weeks among the 31 participants who completed the protocol (77.5% adherence).

The “KEMEPHY” protocol (Ketogenic Mediterranean with Phytoextracts) [70] introduced innovation by incorporating phytotherapeutic extracts and protein preparations designed to mimic traditional Mediterranean foods. Involving 106 participants over 6 weeks, it demonstrated that Mediterranean palatability could mitigate the typical monotony of ketogenic diets, achieving an adjusted adherence rate of 93.4% (excluding dropouts for family/personal reasons unrelated to the dietary protocol). The 10 cm reduction in abdominal circumference suggested preferential effects on visceral fat.

The most ambitious study to date was the Biphasic Ketogenic Mediterranean Diet protocol by Paoli et al. [59], evaluated over a full 12-month period. Its revolutionary design alternated 20-day cycles of ketogenic Mediterranean phases with periods of conventional Mediterranean diet, anticipating the need to address metabolic adaptation through programmed dietary variability. The 88.2% retention rate at 12 months represents an extraordinary achievement in the field of dietary interventions for obesity.

Recent controlled trials have refined our understanding. Cincione et al. [71] demonstrated metabolic superiority of a Mediterranean VLCKD over conventional caloric restriction within just 30 days, with profound improvements in multiple biomarkers. The Keto-Med crossover trial by Gardner et al. [72] provided direct comparative evidence, highlighting differential effects on glycemic control and lipid profile in adults with impaired glucose regulation.

The emerging pattern is both unequivocal and revealing. First, regarding efficacy: weight losses consistently exceed clinically significant thresholds. Second, adherence: with notably high completion rates ranging from 77.5% to 100%, these protocols demonstrate superior tolerability. Investigators attribute this advantage to the greater palatability and nutritional profile of the Mediterranean approach. Common elements include the use of extra virgin olive oil as the primary lipid source (≥30 mL/day), prioritization of fish over red meat, and an abundance of non-starchy vegetables.

However, the final column of Table 6 exposes a universal omission that transforms what might be a celebration of success into an urgent call for innovation: to our knowledge, this review to our knowledge, this is the first systematic analysis suggesting that published protocols have not yet incorporated, measures, or implements strategies to counteract metabolic adaptation.

This critical gap in the literature has profound implications. While these studies demonstrate impressive initial weight losses, none address the inevitable biological phenomenon outlined in Section 4, which compromises long-term maintenance. Evidence from studies such as The Biggest Loser follow-up [63] shows that metabolic adaptation can persist for years, predisposing individuals to weight regain even with reasonable dietary adherence. The absence of strategies to detect and counteract this phenomenon in existing KMD protocols represents a missed opportunity to fully capitalize on the documented metabolic advantage of ketosis.

Therefore, this systematic review not only maps the current state of the field but also illuminates the path toward the next generation of interventions: protocols that integrate the demonstrated efficacy of the KMD with adaptive monitoring and adjustment systems specifically designed to keep metabolism active throughout the entire weight loss and maintenance trajectory. The Adaptive Ketogenic–Mediterranean Protocol (AKMP) represents precisely this conceptual evolution, being the first protocol explicitly designed to fill this critical gap in the literature.

### 7.2. Gaps Identified in the Literature: Opportunities for a New Generation of Protocols

The systematic analysis of the literature, while confirming the efficacy of Ketogenic Mediterranean Diet (KMD) protocols for initial weight loss, also highlights two key opportunities for evolution that define the current frontier of research. These gaps do not invalidate the merits of pioneering studies; rather, they lay the foundation for the next generation of interventions.

The first opportunity lies in the transition from standardized to truly personalized and dynamic design. Existing protocols—including the most sophisticated biphasic model by Paoli et al. [59]—were conceived to apply a uniform intervention in order to demonstrate efficacy under controlled conditions. While this approach was essential for initial scientific validation, it was not designed to adapt in real time to two critical factors:the individual’s life context (including dietary preferences, schedules, and culture), which is key to long-term sustainability; andthe enormous interindividual variability in metabolic response to ketosis, as reflected in weight loss trajectories and lipid changes—a well-documented phenomenon [162].

The logical next step, therefore, is to shift from a predefined “map” to a “GPS” model that continuously adjusts the route.

The second and more profound opportunity is the explicit integration of strategies against metabolic adaptation. Research to date has appropriately focused on demonstrating the primary outcome of weight loss. A consequence of this emphasis is that monitoring of underlying physiological processes, such as adaptive thermogenesis, was not an objective in these studies. However, we now know that the persistence of metabolic adaptation is a key factor in weight regain [163], and that its early magnitude is a predictor of long-term treatment success [164]. The current opportunity lies in synthesizing these two knowledge domains: integrating the insights of metabolic adaptation physiology directly into dietary protocol design.

These two gaps—the absence of dynamic personalization and the omission of anti-adaptive strategies—converge to create a pressing need for a next-generation protocol that transcends the static model. What is needed is an approach that is, by design, biologically intelligent and behaviorally realistic—one that respects not only the physiology of the patient, but also their individuality and daily life.

### 7.3. Design of the Adaptive Ketogenic–Mediterranean Protocol (AKMP): Principles and Phases

The Adaptive Ketogenic–Mediterranean Protocol (AKMP) constitutes an adaptive therapeutic system that transcends the limitations of the static protocols identified in the systematic review. Its structure integrates, for the first time, a dynamic biomarker-based adjustment mechanism specifically designed to actively counteract metabolic adaptation.

#### 7.3.1. Foundational Principles

The AKMP is grounded in three cardinal principles that define its therapeutic identity:Principle 1: Optimized Ketogenic–Mediterranean Synergy.The protocol uses the metabolic potency of nutritional ketosis with the cardioprotective profile of the Mediterranean pattern. This integration includes extra virgin olive oil (30–50 mL/day) as the main lipid source, oily fish (≥4 servings/week) to optimize the omega-6:omega-3 ratio (~4:1)—a proportion shown to yield significant benefits in secondary cardiovascular prevention [165]—and the maximum allowable quantity of vegetables, limited to a daily cap of 20 g of net carbohydrates.Paoli et al. have demonstrated that converging the ketogenic and Mediterranean patterns improves both adherence and cardiometabolic risk profile [166]. In an independent study, Paoli’s biphasic protocol achieved sustained weight loss of 16.9 ± 7.1 kg with no regain during 6 months of Mediterranean maintenance, and an adherence rate of 88.2% [59].Principle 2: Systematic Exploitation of the Metabolic Advantage.The AKMP is designed to maximize the increase in energy expenditure documented under nutritional ketosis. Fine and Feinman proposed that the “metabolic advantage” of low-carbohydrate diets reflects real differences in metabolic pathway efficiency, without violating the laws of thermodynamics [85,158].This advantage—typically in the ≈100–300 kcal/day range and context-dependent, as detailed in Section 6—derives from multiple mechanisms, including the high thermic effect of protein, ATP-dependent futile cycles, and the energetic cost of gluconeogenesis. The Framingham State Food Study confirmed that participants on a low-carbohydrate diet experienced a ~210 kcal/day increase in total energy expenditure compared to a high-carbohydrate diet [83]. The meta-analysis by Ludwig et al. showed an increase of 50.4 kcal/day for every 10% carbohydrate reduction in studies lasting more than 2.5 weeks [84].Principle 3: Dynamic Anti-Stall Personalization.Unlike all previously published protocols, the AKMP implements a system of continuous monitoring and proactive adjustments. Weight plateaus, operationally defined here as a variation < 1% over 14 days with confirmed ketosis—based on the work of Heymsfield et al. [167], which demonstrated weight stability with a coefficient of variation < 0.5% over 10 days, extended to 14 days for greater clinical certainty—trigger personalized interventions:strategic protein increase (up to 1.6 g/kg ideal body weight) or modest caloric reduction (100–200 kcal/day), guided by clinical dialogue concerning satiety and individual patient preferences.

#### 7.3.2. Temporal Architecture and Nutritional Principles of the Protocol

The Adaptive Ketogenic–Mediterranean Protocol (AKMP) is structured into three sequential phases designed to maximize the metabolic advantage of ketosis while facilitating the transition to a sustainable Mediterranean dietary pattern. Unlike all previously published KMD protocols, which maintain static caloric and macronutrient levels, the AKMP is the first to recognize and implement specific strategies to counteract adaptive thermogenesis through dynamic adjustments based on individual metabolic response. Each phase targets specific physiological and behavioural objectives, implemented under the continuous monitoring system detailed in Section 7.4.

Phase 1: Ketogenic–Mediterranean Induction (2 weeks)

The 14-day duration is based on multiple physiological and behavioural considerations. This period allows for the capture of initial water loss associated with glycogen depletion, which varies depending on individual reserves. Simultaneously, it provides sufficient time for ketone bodies to exert their documented anorexigenic effect [168,169,170], facilitating the transition from potentially addictive dietary patterns rich in refined carbohydrates to nutritional ketosis. Visible weight loss during these first weeks reinforces initial motivation, a critical success factor for long-term outcomes according to established literature [54].

Nutritional principles:Carbohydrates: ≤20 g net/day;Protein: 1.2–1.6 g/kg ideal body weight (20–30% of total energy intake);Fats: 60–70% of total energy intake;Hydration: ≥2 L/day.

Personalized Mediterranean design:

Menus are individualized using Dietopro® software (accessed on 12 June 2025), accounting for multiple factors: food preferences, work schedules, allergies and intolerances, budget, cooking skills, local food availability, and cultural or religious considerations. Each menu includes weekly meals with exact quantities and preparation suggestions, prioritizing:Extra virgin olive oil: 30–40 mL/day as the primary lipid source;Mediterranean oily fish: ≥4 servings/week;Non-starchy vegetables, customized per individual plan;Mediterranean nuts, within carbohydrate limits.

When the design fails to meet Recommended Daily Intakes, multivitamin supplementation is prescribed or any nutritional deficit is addressed in the next menu cycle.

Phase 2: Adaptive Metabolic Optimization (individualized duration)

This phase is the operational core of the protocol, maintaining nutritional ketosis while activating the dynamic adjustment system to counteract metabolic adaptation. Nutritional principles remain similar to Phase 1, with personalized modifications based on individual response as detected through continuous monitoring. Specifically, upon detection of weight plateaus (defined as <1% variation over 14 days with confirmed ketosis), individualized adjustments are implemented:Increased protein intake if persistent hunger is reported (up to 1.6 g/kg);Modest caloric reduction (100–200 kcal/day) if satiety is adequate (as detailed in Section 7.4).

The primary objective is to harness the documented metabolic advantage of ketosis (+100–300 kcal/day) [83,84] to prolong periods of active weight loss, implementing proactive adjustments upon detection of metabolic plateaus.

Phase 3: Antioxidant Transition and Mediterranean Consolidation (4 weeks)

This phase represents a key innovation in the AKMP, designed to address the metabolic changes that occur during the transition from ketosis to a maintenance-oriented dietary pattern. The strategic reintroduction of carbohydrates, with an emphasis on high-antioxidant foods, is structured to meet multiple physiological objectives.


*Biochemical Justification for the Antioxidant Emphasis:*


The shift in energy substrate from ketone bodies to glucose entails a mitochondrial metabolic reorganization that may transiently increase the production of reactive oxygen species (ROS). During prolonged ketosis, mitochondria undergo a profound restructuring of their enzymatic machinery, with reported twofold increases in the activity of Complex I (NDUFB8 subunit) and Complex IV (MTCO1), alongside upregulation of PDK4, which reduces pyruvate oxidation by ~30%, while the fatty acid oxidation capacity doubles [171,172,173]. Although no studies have directly measured ROS production during the transition from ketosis to glycolytic metabolism, it is biologically plausible that this substrate shift may induce transient oxidative stress, based on the established concept of metabolic flexibility [174].

This hypothesis of transient oxidative stress becomes more relevant when considering the complex modulation of cellular defenses. During ketosis, endogenous metabolites such as β-hydroxybutyrate exert potent signaling effects, including histone deacetylase (HDAC) inhibition, which modulates gene expression [160]. Simultaneously, the Nrf2/ARE system—recognized as the master regulator of the cellular antioxidant response [175]—is a key target of phytochemicals present in the Mediterranean diet.

The confluence of these factors—a metabolic state with active signaling and a diet-sensitive defense system—suggests that profound metabolic transitions, such as the shift from ketosis to glycolysis, represent a window of vulnerability in which proactive reinforcement of antioxidant systems constitutes a physiologically coherent strategy.

We hypothesize that selected dietary antioxidants may act synergistically to modulate this transition. Polyphenols from extra virgin olive oil (e.g., hydroxytyrosol, oleuropein) have shown the ability to reduce lipid peroxidation and protect mitochondrial function [176]. Anthocyanins from berries exert protective effects on vascular endothelium, particularly relevant during metabolic transitions [177]. Isothiocyanate derivatives from cruciferous vegetables, such as sulforaphane, are recognized as potent activators of Nrf2, the master regulator of the cellular antioxidant response [175].

Carbohydrate Reintroduction Timeline:

Week 1: 50–75 g carbohydrates/day

Priority given to non-starchy vegetables rich in phytochemicals:
○Cruciferous vegetables (sulforaphane);○Leafy greens (lutein, zeaxanthin);○Colorful vegetables, such as red and yellow peppers (β-carotene) and tomatoes (lycopene).
Gradual introduction of berries: offering maximum antioxidant density with minimal glycaemic impact.

Week 2: 75–100 g carbohydrates/day

Incorporation of legumes in controlled portions: a source of complex carbohydrates, plant-based protein, and bioactive compounds (e.g., saponins, phytates with antioxidant properties).Enhancement with Mediterranean spices:
○Turmeric (curcumin);○Oregano (rosmarinic acid);○Rosemary (carnosol).
Maintenance of extra virgin olive oil intake: 40–50 mL/day.

Weeks 3–4: Consolidation of the Mediterranean Pattern

Monitoring of weight stabilization, taking into account the recovery of hepatic and muscle glycogen stores.Establishment of a modified Mediterranean dietary pattern, integrating the metabolic insights gained during the ketogenic phase.

This strategic transition not only facilitates long-term sustainability but also capitalizes on the well-documented cardiovascular benefits of the Mediterranean pattern [59,69,70], thus creating a bridge between intensive therapeutic intervention and a permanent healthy lifestyle.

This approach—which combines the gradual reintroduction of carbohydrates with antioxidant-rich foods, while maintaining residual levels of ketone bodies with documented antioxidant properties [160]—represents an innovative strategy to facilitate metabolic substrate switching.

#### 7.3.3. Methodology for Energy and Macronutrient Personalization

To ensure a scientifically grounded and individualized intervention from the outset, the AKMP employs a stratified approach to determine energy requirements.

The first step is the estimation of Resting Energy Expenditure (REE) using the Mifflin-St Jeor equation, which is currently considered the most accurate formula in clinical practice across a wide range of populations [178]:Men: REE = (10 × weight in kg) + (6.25 × height in cm) − (5 × age in years) + 5Women: REE = (10 × weight in kg) + (6.25 × height in cm) − (5 × age in years) − 161

To ensure maximum accuracy across varying degrees of obesity, the calculation is stratified as follows:For participants with BMI < 40 kg/m^2^: actual body weight is used in the equation.For participants with BMI ≥ 40 kg/m^2^: an adjusted body weight is used to avoid REE overestimation due to excess fat mass. The formula applied is a standard in clinical dietetics [179]:
○Adjusted Weight = Ideal Weight + 0.25 × (Actual Weight − Ideal Weight)


Once REE is obtained, total energy expenditure (TEE) for maintenance is calculated by multiplying REE by a Physical Activity Factor (PAF). Although the Institute of Medicine (IOM) reference report defines conceptual Physical Activity Level (PAL) categories through complex regression models [180], the AKMP adopts simplified clinical multipliers derived from those PAL categories:PAF 1.2: Sedentary lifestyle (corresponding to PAL 1.0–1.39);PAF 1.375: Light activity (corresponding to PAL 1.4–1.59);PAF 1.55: Moderate activity (corresponding to PAL 1.6–1.89);PAF 1.725: Intense or very active lifestyle (corresponding to PAL 1.9–2.5).

This initial maintenance TEE provides the foundation for designing the Phase 1 meal plan, establishing an individualized caloric deficit that is compatible with the protocol’s macronutrient targets and supports a safe and effective induction into nutritional ketosis.

### 7.4. Dynamic Adjustment System to Optimize Weight Loss

Sustained weight loss remains one of the greatest challenges in obesity management, with long-term weight regain rates exceeding 80% within 5 years [54]. While no dietary protocol can indefinitely prevent weight plateaus—a biologically inevitable feature of metabolic adaptation—the AKMP implements a dynamic monitoring and adjustment system designed to detect plateaus early and respond proactively, thereby potentially extending periods of active weight loss by leveraging the documented ketogenic metabolic advantage of ≈100–300 kcal/day, which is context-dependent [83,84].

Monitoring and Adjustment System of the AKMP

The protocol operates via a continuous cycle of monitoring, analysis, and adjustment, allowing the intervention to be adapted to the individual patient’s response (Figure 7).

Systematic Monitoring:A stratified surveillance system is implemented. For high-frequency tracking, twice-weekly measurements of fasted body weight are combined with urine ketone testing using Ketostix^®^ strips—a well-documented, practical, low-cost tool for adherence monitoring [181].In the AKMP, ketone measurements are obtained from the first morning urine sample to ensure standardization with fasted weight measurements and maximize patient convenience. Every two weeks, capillary ketonemia (β-hydroxybutyrate) is quantified, with nutritional ketosis defined as 0.6–3.0 mmol/L [166] (In formal research pilots (Section 8.3.2), biomarker frequency is temporarily intensified as per the study schedule.). Bioimpedance analysis (InBody 270) is used when hydration-related changes are suspected of masking actual fat loss [182]. Clinical-practice (pragmatic) versus research-grade monitoring. In routine care, we prioritize low-burden tools—fasted body weight (≥2×/week), weekly urine acetoacetate strips for adherence checks (with explicit acknowledgment of their limitations due to hydration and diurnal variation), and capillary β-hydroxybutyrate at least every two weeks or whenever an adjustment is considered. As cost-efficient “triggers,” consecutive negative urine strips or an abrupt weekly weight uptick (compatible with glycogen/water repletion) prompt an earlier in-clinic capillary β-OHB check. In research settings, gold-standard assessments include body composition by DXA (with operational BIA validated against DXA in a pilot), indirect calorimetry for REE, accelerometry for activity, and DLW for TEE in validation subsets.Plateau Detection:A weight plateau is operationally defined as <1% body weight variation over 14 consecutive days with confirmed nutritional ketosis. This 14-day period allows differentiation between normal daily fluctuations in body weight—typically 1–2 kg due to hydration status, glycogen stores, and gastrointestinal contents—and true stagnation in fat loss [183]. When a plateau is detected, an evaluation visit is scheduled to:
○Confirm adherence through ketonemia ≥ 0.6 mmol/L;○Analyze body composition;○Clinically assess the patient’s general status.The 14-day plateau window is a pragmatic choice that balances day-to-day weight variability (fluid/glycogen/intestinal contents) with clinical feasibility; future testing will examine sensitivity to 10- versus 14- versus 21-day windows.Adjustment Strategies:Upon confirmed plateau, one of two strategies is implemented based on clinical evaluation:
○Protein Adjustment:If the patient reports persistent hunger despite confirmed ketosis, protein intake is increased incrementally up to 1.6 g/kg ideal body weight. This strategy leverages protein’s higher thermic effect (20–30% of energy content versus 5–10% for carbohydrates and 0–3% for fats) [157], potentially helping to maintain total energy expenditure while enhancing satiety [184].○Caloric Adjustment:If the patient reports adequate satiety, a modest caloric reduction of 100–200 kcal/day is introduced, primarily by reducing fat intake while preserving protein.

The choice between strategies is guided by clinical dialogue, considering not only hunger but also patient preferences, prior adherence, and life context. Following implementation of an adjustment, the modified protocol is maintained for a minimum of 14 days—a period established to allow detection of stable metabolic responses beyond transient fluid and glycogen fluctuations—before evaluating the need for further modifications.

Advantages and Limitations of the Adaptive Approach

It is essential to recognize that this system does not prevent plateaus—which are inherent to any weight loss process—but instead offers specific advantages:Early plateau detection enables timely intervention before motivation and adherence deteriorate.Individualized response strategies are tailored to each patient’s specific metabolic response and subjective experience, rather than applying generic modifications.Strategic utilization of the ketogenic metabolic advantage, which increases total energy expenditure by 100–300 kcal/day [83,84], may extend periods of active weight loss when combined with timely adjustments.

This adaptive approach distinguishes the AKMP from static protocols that maintain fixed macronutrient targets regardless of individual response. While GLP-1 receptor agonists act primarily through appetite suppression without directly addressing metabolic adaptation [80], the AKMP aims to optimize both adherence and energy expenditure through biomarker-informed adjustments, acknowledging that weight plateaus are an inevitable component of the weight loss process that can be proactively managed rather than passively accepted.

Figure 7 visually depicts this proactive management system, providing clinicians with a clear decision-making algorithm for implementing the dynamic adjustments that define the AKMP’s innovative approach to metabolic adaptation.

### 7.5. Comparative Analysis: Positioning the AKMP Within Current Therapeutic Paradigms

Having established the foundational principles of the AKMP and its dynamic adjustment system, a critical question arises: How does this proposal compare to the dominant therapeutic paradigms?

This comparative analysis demonstrates that the AKMP represents the first intervention explicitly designed to simultaneously address the two fundamental barriers identified throughout this review:Adherence compromised by neurobiological factors, addressed through its culturally acceptable Mediterranean foundation;Inevitable metabolic adaptation, countered through the systematic exploitation of the ketogenic metabolic advantage (100–300 kcal/day).

While current paradigms either address these challenges partially or neglect them entirely, the AKMP integrates—at least conceptually—specific mechanisms for each barrier. Table 7 presents this comparative matrix.

Analysis of this matrix reveals limitations within each paradigm. Conventional hypocaloric diets exhibit an average long-term adherence of only 60.5% [136] and result in a loss of lean mass equivalent to approximately one-quarter of total weight loss, compromising basal metabolism [185].

GLP-1 receptor agonists, while revolutionary, impose pharmacological dependence [80]. Critically, in terms of body composition, studies have consistently documented that, alongside fat loss, there is also a significant reduction in lean mass [186,192]. Moreover, their efficacy relies solely on appetite suppression, without any direct mechanism to counteract the decline in energy expenditure [81,190].

Standard ketogenic diets, while metabolically advantageous [83,84], often lack the structure and palatability required to ensure sustainability, a feature that KMD protocols have demonstrated in the short term [70].

A key finding from the DIRECT study established that initial weight loss at 6 months is the strongest predictor of long-term maintenance success [191], underscoring the importance of optimizing early intervention—a central design principle of the AKMP’s dynamic structure.

The AKMP, as a theoretical construct, aims to synthesize the strengths of each paradigm while implementing innovations to address their limitations. It is important to acknowledge the potential limitations of the AKMP itself: its implementation would require substantial resources and a high level of patient engagement, which may limit its scalability. These practical considerations must be addressed in future feasibility studies.

### 7.6. Ethical and Safety Considerations for AKMP

Although this is a review, the AKMP proposes an adaptive nutritional protocol that entails biomarker monitoring and macronutrient adjustments. Clinical implementation must therefore follow standard ethical safeguards: (i) explicit informed consent covering the rationale, potential benefits, and foreseeable risks; (ii) supervision by clinicians trained in ketogenic therapy; (iii) adherence to absolute contraindications and precautions summarized in Section 3.2.5; (iv) a pragmatic, low-burden monitoring schedule as detailed in Section 7.4; and (v) specific caution with SGLT-2 inhibitors owing to the risk of euglycemic ketoacidosis during ketogenic phases, with temporary discontinuation when appropriate. Special populations—type 1 diabetes, pregnancy/lactation, advanced CKD, hepatic insufficiency—require individualized risk–benefit assessment and, where necessary, exclusion. Any future clinical evaluation of the AKMP will require prior ethics approval and safety reporting consistent with Good Clinical Practice.

## 8. General Discussion, Limitations, and Future Directions

To maintain conceptual continuity with the Introduction while avoiding duplication, this General Discussion frames obesity as a “perfect storm” arising from the synergistic convergence of a pro-inflammatory, high-glycemic food supply, an obesogenic environment, and heterogeneous biological susceptibility. Against this backdrop, we focus on aspects not previously detailed: why static protocols fail within this storm; how GLP-1 receptor agonists relieve one pressure (appetite) yet leave the suppression of energy expenditure untouched; and how the AKMP is designed as an anti-adaptive, biomarker-guided dietary architecture to withstand—and gradually dissipate—that storm.

### 8.1. Synthesis of the Central Argument: From Metabolic Adaptation to Adaptive Solution

This study articulates a systematic argument identifying two fundamental obstacles in the long-term management of obesity. The first—and most significant—is treatment adherence, which represents the greatest challenge in real-world settings [59,60]. However, the modern understanding of this issue transcends traditional explanations such as “lack of willpower” or “insufficient motivation.” Emerging neurobiological evidence indicates that, in 10–25% of people with obesity, poor adherence may be mediated by food-addiction mechanisms. Ultra-processed foods rich in refined carbohydrates activate dopaminergic reward circuits comparable to those triggered by drugs of abuse [29], generating intense cravings, loss of control, and relapse patterns that sabotage even the sincerest attempts at dietary change.

This neurobiological perspective explains why purely educational or motivational interventions repeatedly fail in certain individuals: they are not battling a mere food preference, but rather a real neurochemical dysregulation of their reward systems.

The second obstacle—equally critical but less widely recognized—is metabolic adaptation [61], a powerful physiological response that the body activates following weight loss. This phenomenon, characterized by a suppression of thermogenesis up to 15% beyond what would be expected from changes in body composition [62], along with hormonal alterations that promote weight regain, was dramatically illustrated in the 6-year follow-up of participants from The Biggest Loser [63]. Taken together, the convergence of a high-glycemic, ultra-processed food environment with neurobiological vulnerability, chronic sedentarism, and the persistence of metabolic adaptation constitutes a “perfect storm” that overwhelms homeostatic weight regulation and renders static interventions fragile over the long term.

The critical analysis of current therapeutic paradigms reveals fundamental limitations on both fronts. Static ketogenic diets, despite their initial efficacy and documented metabolic advantage of 100–300 kcal/day [83,84], lack structured mechanisms to support long-term adherence in real-life settings—especially among patients with neurobiological alterations in reward circuitry—and to adapt to weight plateaus [59,60].

GLP-1/GIP receptor agonists, while revolutionary in their ability to induce 15–22% weight loss [78,79], act through pharmacological appetite suppression without counteracting the decline in energy expenditure. They require indefinite administration and entail costs that are inaccessible to many, and are associated with weight regain of approximately two-thirds of the weight initially lost after discontinuation [80].

On this basis, the Adaptive Ketogenic–Mediterranean Protocol (AKMP) emerges as a theoretical proposal specifically designed to simultaneously address both challenges through:Integration of the Mediterranean patterns to enhance adherence and palatability;Exploitation of the metabolic advantage of ketosis;Personalized adjustments based on individual response to counteract metabolic adaptation.

### 8.2. The AKMP as a Proposal for Personalized Nutritional Medicine

The innovation of the Adaptive Ketogenic–Mediterranean Protocol (AKMP) lies in its individualized and adaptive approach. Rather than applying a rigid, one-size-fits-all protocol, the AKMP recognizes that each patient has unique needs based on their metabolic history, dietary preferences, and life context. This personalization is grounded in the continuous monitoring of both objective markers (ketonemia, body composition) and subjective indicators (satiety, energy levels), which enable the protocol to be adjusted according to individual response, as described in detail in Section 7.

The vision of the AKMP extends beyond obesity management. Its core principles—glycemic control through nutritional ketosis, preservation of lean mass, and response-based personalization—have potential applications in type 2 diabetes, metabolic syndrome, and healthy aging, conditions previously addressed in earlier chapters.

The evidence on the anti-inflammatory effects of β-hydroxybutyrate, presented in Section 6, even suggests possible applications in broader inflammatory contexts.

### 8.3. Integration with Pharmacotherapy: Acknowledging Patient Diversity

It is essential to recognize that there is no single optimal strategy for all patients. The approach must be personalized according to individual context:For patients with mild-to-moderate obesity and strong motivation: A well-calibrated, balanced diet supervised by a professional, combined with increased physical activity, may suffice as first-line therapy.For patients with capacity and motivation for exercise: Beginning with a structured physical activity program may be ideal, with dietary modifications added as needed.For motivated individuals without severe carbohydrate addiction: They may start directly with the AKMP, without requiring pharmacological support, benefiting from the natural appetite suppression induced by ketosis.For severe cases or patients with strong addiction to refined carbohydrates: Recognizing the neurobiological basis of this phenomenon [25], GLP-1 receptor agonists may constitute an essential first-line therapeutic tool, always combined with nutritional supervision to minimize metabolic adaptation and the loss of muscle mass characteristic of conventional hypocaloric diets.

However, relying solely on pharmacological appetite suppression presents fundamental limitations. Given that certain patterns of compulsive consumption of ultra-processed foods involve a dysregulation of the dopaminergic reward pathway [25], GLP-1 agonists may serve to temporarily correct altered hunger and satiety signals, creating a therapeutic window of opportunity.

But the ultimate goal cannot be indefinite pharmacological dependence. True therapy begins when that window is used to implement a strategy like the AKMP, which aims to retrain both metabolic and behavioral responses, thus fostering lasting metabolic resilience and dietary autonomy.

This understanding underpins our proposal for an integrated strategy: the AKMP as a concomitant intervention alongside pharmacotherapy, including highly effective dual agonists such as tirzepatide. The AKMP is specifically designed to prepare patients for eventual drug discontinuation by establishing metabolically advantageous and behaviorally sustainable eating patterns [83,84]. Crucially, when combined with potent incretin agonists, the AKMP aims to optimize the quality of weight loss—mitigating lean-mass loss via adequate protein intake and the anti-catabolic milieu of ketosis—and to counter the metabolic neutrality of these agents by leveraging the ketogenic metabolic advantage [83,84,153].

Furthermore, the personalized framework of the AKMP allows integration of synergistic strategies for specific subpopulations. Post-menopausal women face unique metabolic and vascular challenges that complicate weight maintenance and elevate NCD risk. In this context, combining the AKMP with regular physical activity and targeted nutritional components such as soy isoflavones may provide complementary benefits. Prospective cohort evidence links higher isoflavone intake—particularly tofu—with lower coronary heart disease risk [193], while mechanistic and clinical data support metabolic and vascular effects within menopausal care [194]. Together, these findings suggest that, in selected post-menopausal patients, adjunctive soy isoflavones could be considered alongside AKMP to optimize cardiometabolic outcomes.

#### 8.3.1. Sequential Therapeutic Algorithms: AKMP-First (Algorithm 1) vs. Incretin-First Pathways (Algorithm 2)

Two pragmatic sequences can reconcile biological efficacy with real-world adherence. First, a Ketogenic–Mediterranean lead-in (AKMP-first) can produce rapid, motivating weight loss and attenuate appetite via ketosis, potentially “recalibrating” reward-related eating by stabilizing glycaemia and lowering ghrelin [26,169,195]. Second, when initial adherence to carbohydrate restriction is unlikely, an incretin-first approach can pharmacologically dampen hunger and reward drive, creating a therapeutic window to progressively transition into AKMP [145,196].
**Algorithm 1. AKMP-first, step-up to incretin therapy if adherence threatens to fail**Indication: motivated patients willing to adopt structured menus and monitoring.Steps: (i) Initiate ketogenic–Mediterranean induction per AKMP Phase 1; (ii) maintain dynamic, biomarker-guided adjustments (protein up to 1.6 g/kg IBW if hunger; or −100–200 kcal/day if satiety adequate); (iii) Add a GLP-1/GIP agonist if any of the following triggers occur:
–confirmed plateau ≥ 14 days with ketosis plus rising hunger (VAS ≥ 60/100) despite protein optimization;–adherence hazards: social pressure/events, family meal constraints, travel, cost/time burden of meal prep, taste monotony, GI symptoms (constipation), seasonal festivities, caregiver stress;–repeated ketone < 0.3 mmol/L despite counseling;–high risk neurobehavioral patterns (food-addiction phenotype) jeopardizing continuation
Mechanistic aim: preserve lean mass (adequate protein + anti-catabolic ketosis) and offset incretin metabolic neutrality by leveraging the ketogenic metabolic advantage [79,83,153], while dual agonism (tirzepatide) may permit dose escalation via GIP-mediated nausea attenuation and enhance efficacy [79,147,197].


**Algorithm 2. Incretin-first, tapered carbohydrate reduction to AKMP**
Indication: patients unable to “on-ramp” into AKMP due to reward-driven eating, severe cravings, chaotic schedules, or prior failed ketogenic attempts.Steps: (i) Start a professionally supervised Mediterranean diet; (ii) initiate a GLP-1RA or dual GLP-1/GIP agonist; (iii) taper carbohydrates by 25–50 g every 2 weeks while monitoring satiety and biomarkers (target β-hydroxybutyrate ≥ 0.6 mmol/L for ≥2 consecutive weeks); (iv) when hunger/food preoccupation is blunted and BHB threshold is met, transition to AKMP Phase 2 (adaptive metabolic optimization).Mechanistic aim: use incretin therapy to reduce appetite and food-reward responsivity [145,196], then consolidate the behavioral and metabolic gains under AKMP’s biomarker-guided framework.

Under energy- and protein-matched conditions, long-term weight-loss differences between macronutrient distributions may be small [154,155,156]. The AKMP therefore emphasizes *phase-appropriate* synergy: (i) leverage ketosis for appetite control and potential TEE gains in free-living contexts; (ii) recruit incretin pharmacotherapy when adherence falters or as an on-ramp toward sustained carbohydrate restriction.

#### 8.3.2. Pragmatic Pilot Trial to Validate the AKMP–Incretin Sequencing

As a pragmatic step toward empirical validation of the sequential pathways in Section 8.3.1, we propose a 24-week, four-arm randomized pilot designed primarily to establish feasibility and adherence, while providing variance estimates for future trials. Participants (adults with obesity, with or without stable prediabetes/T2D) would be allocated 1:1:1:1 in permuted blocks, stratified by sex and T2D status, to: (A) usual care with standard non-ketogenic dietary advice; (B) tirzepatide alone with general non-ketogenic guidance; (C) AKMP→tirzepatide (weeks 0–12 AKMP—including the dynamic anti-plateau adjustments—and weeks 12–24 addition of tirzepatide while maintaining AKMP or transitioning to Mediterranean maintenance per response); or (D) tirzepatide→AKMP (weeks 0–12 tirzepatide with non-ketogenic Mediterranean counseling, followed by a structured transition to AKMP during weeks 12–24). Outcome assessors for the optional calorimetry/body-composition substudies would remain blinded to group assignment.

Monitoring follows the pragmatic framework in Section 7.4 (twice-weekly fasted body weight; first-morning urine ketone strips; bi-weekly capillary β-hydroxybutyrate; bioimpedance when fluid shifts are suspected), with a trial-specific intensification during induction (daily or alternate-day BHB in weeks 1–2, then 2–3 times/week thereafter; urine strips as supportive evidence). A wrist accelerometer in a subsample will capture NEAT as an exploratory mechanistic measure.

Schedule and measurements are anchored to metabolic-adaptation biology: indirect calorimetry for resting energy expenditure (REE) and DXA fat-free mass at weeks 0, 12, and 24, enabling calculation of residual adaptive thermogenesis (observed REE minus body-composition-predicted REE). Fasting lipids (including subfractionation by NMR where available) will be collected at baseline and week 24. Safety surveillance includes standardized algorithms for antihypertensive and glucose-lowering medication adjustments; type 1 diabetes is excluded; concurrent SGLT-2 inhibitors are either exclusionary or require suspension before ketogenic phases to mitigate euglycemic ketoacidosis risk (consistent with Section 3.2.5).

Operational plateau handling adheres to Section 7.4: <1% weight variation over 14 consecutive days with confirmed nutritional ketosis triggers the AKMP adjustment algorithm (protein up-titration when hungry; modest fat-calorie reduction when satiety is adequate), maintained for ≥14 days before re-evaluation. To inform later trials without adding participant burden, sensitivity analyses with 10- and 21-day moving windows could also be explored post hoc.Primary readouts are feasibility/adherence metrics: retention to week 24; compliance with paired weight-and-ketone reports (target ≥ 75%); and, in AKMP arms, the proportion of weeks with verified ketosis. Early efficacy measures include percent weight change at 12 and 24 weeks, waist circumference, proportions achieving ≥5% and ≥10% loss, HbA1c in prediabetes/T2D, and safety events (GI symptoms, hypotension, hypoglycemia, eDKA signals). A total sample of ~64–80 (16–20 per arm) yields 95% CIs of approximately ±10–12 percentage points around an 80% retention rate, sufficient for feasibility judgments and variance estimation. Analyses will follow intention-to-treat, using linear mixed-effects models for weight trajectories (time, arm, and time×arm), Cox models for time-to-first plateau, and an exploratory per-protocol analysis (≥70% reporting adherence and prespecified menu adherence) to probe mechanistic signals. This proposed pilot is conceptual and has not been conducted; it operationalizes validation strategies while preserving the non-experimental nature of the present review.

*Note to the reader:* This subsection details a pragmatic pilot framework intended to dimension and refine subsequent controlled trials comparing AKMP-first versus incretin-first sequences within routine clinical constraints.

### 8.4. Limitations and Future Directions

We must fully acknowledge the limitations of this work with complete transparency. First, our evidence-mapping component was restricted to English-language publications, which introduces a risk of language restriction bias. Second, we did not include grey literature or non-peer-reviewed sources; while this improves specificity, it may exclude relevant protocols. Third, definitions and naming conventions for ‘Ketogenic–Mediterranean Diet’ vary across studies (e.g., macronutrient thresholds, phase structure, and ketosis verification), resulting in heterogeneity that limits direct comparability. We mitigate these issues by (i) explicitly reporting search strategies and selection criteria, (ii) providing study-level details and quality ratings in Appendix A, and (iii) discussing implications of definitional heterogeneity in the Results.

The AKMP is currently a theoretical proposal, grounded in solid physiological principles, but lacking direct empirical validation. Its practical implementation faces several significant challenges:Requires specialized personnel in ketogenic nutrition and metabolic monitoring;Necessitates infrastructure for biomarker tracking;Demands a high level of patient commitment;Its complexity limits scalability in primary care settings.

Uncertainties remain regarding key aspects such as the optimal duration of ketogenic phases, the long-term impact on lipid profiles, and the sustainability of benefits following multiple dietary transitions. Furthermore, the extrapolation of the ketogenic metabolic advantage to repeated dynamic adjustments requires specific validation.

The research agenda is clear:Pilot studies (3–6 months) to assess feasibility and refine the protocol;Randomized trial (12 months) comparing AKMP vs. static Ketogenic Mediterranean Diet, evaluating weight loss, body composition, and metabolic markers;Comparative trial (24 months) AKMP vs. semaglutide, measuring not only weight loss but also its quality (lean mass/fat ratio), energy expenditure, and post-intervention maintenance;Combination studies assessing the proposed synergy between pharmacotherapy and the AKMP.

### 8.5. Final Reflection: Personalizing the Approach to Obesity

This manuscript argues that successful obesity management requires moving beyond the “one-size-fits-all” paradigm and adopting a truly personalized approach. There are no universally superior strategies; the optimal intervention must consider the biological, psychological, and social context of each individual.

The modern environment presents unprecedented challenges: chronic stress, extreme sedentarism, and foods engineered to maximize consumption. Traditional recommendations—though valuable—are often insufficient in the face of this reality. In this context, more intensive interventions such as the AKMP may offer the tools needed to attempt a more durable metabolic recalibration.

The future of obesity treatment lies not in choosing between diet and pharmacotherapy but in their intelligent, personalized integration. Every patient deserves an approach that recognizes their unique situation and provides the tools—whether dietary, pharmacological, behavioral, or combined—that are most appropriate for achieving and maintaining a healthy weight. The AKMP represents a step toward a future of truly personalized nutritional medicine.

## 9. Conclusions

Metabolic adaptation constitutes an inevitable physiological response to weight loss—a fundamental biological challenge that no current therapeutic paradigm, whether dietary or pharmacological, has yet succeeded in either preventing or sustainably reversing. This reality, exacerbated by a modern food environment dominated by ultra-processed products that can trigger compulsive eating patterns, exposes the fundamental incompatibility between static interventions and the inherently dynamic nature of human metabolism.

The future of obesity management lies in adaptive therapeutic architectures. The Adaptive Ketogenic–Mediterranean Protocol (AKMP) proposed herein exemplifies this concept by synergistically integrating the metabolic advantages of nutritional ketosis—including its an estimated thermogenic effect (≈100–300 kcal/day), which is context-dependent and its potential to modulate reward circuitry—with dynamic personalization systems that respect each patient’s biology, life context, and cultural preferences.

The empirical validation of such adaptive models represents not merely the next logical step in obesity research, but an imperative to translate decades of metabolic science into clinically accessible solutions, marking a paradigm shift from chronic symptomatic control to truly personalized nutritional medicine.

By explicitly pairing real-time metabolic monitoring with culturally rooted dietary practices, the AKMP offers a previously untested, patient-centerd framework that warrants rigorous clinical evaluation.

## Figures and Tables

**Figure 1 nutrients-17-02699-f001:**
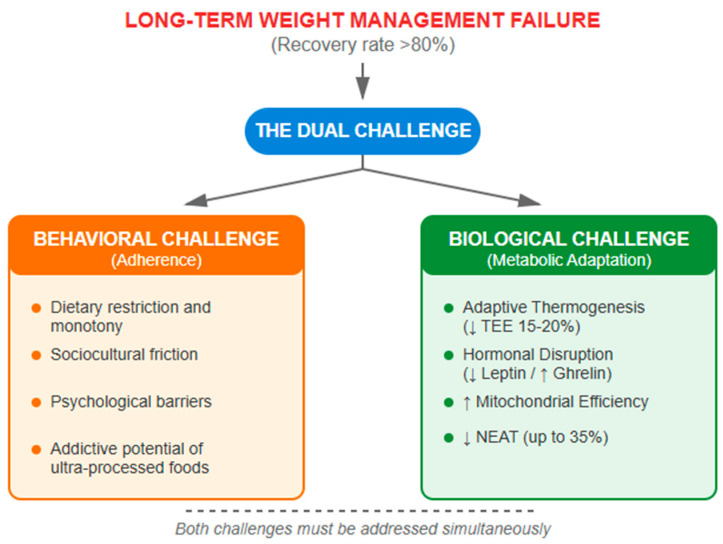
The dual challenge in long-term obesity management. The failure to maintain weight loss (>80% recovery rate) [54] results from the convergence of two fundamental challenges. The behavioral challenge encompasses barriers such as the addictive potential of ultra-processed foods [25]. The biological challenge comprises metabolic adaptation [61], characterized by adaptive thermogenesis (a reduction in total energy expenditure (TEE), that is 15–20% greater than predicted by changes in body composition alone) [62], hormonal disruptions (decreased leptin/increased ghrelin) [64,65], enhanced mitochondrial efficiency [66,67], and reduced non-exercise activity thermogenesis (NEAT) [68]. Successful long-term weight management must address both challenges simultaneously.

**Figure 2 nutrients-17-02699-f002:**
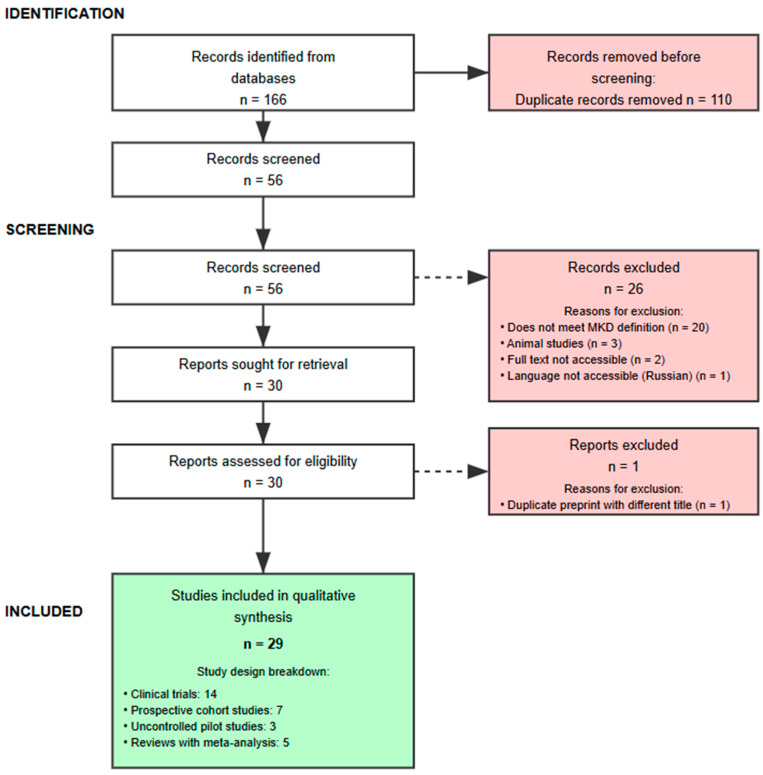
Flow diagram to select studies, according to aim of this manuscript.

**Figure 3 nutrients-17-02699-f003:**
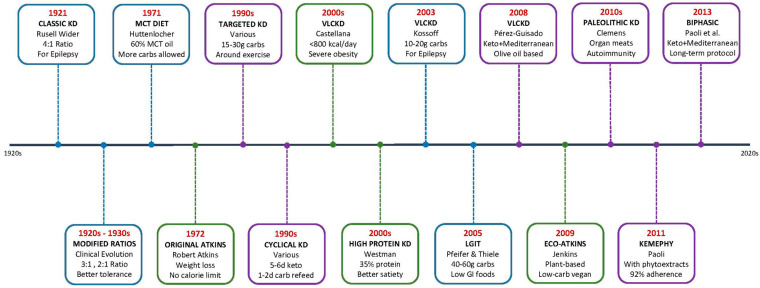
The timeline illustrates the progression from the classical ketogenic diet toward increasingly flexible and specific variants. Key milestones include innovations in composition (MCT, modified ratios), expanded applications (obesity, athletic performance), and convergence with established dietary patterns (Mediterranean, vegetarian). Note the acceleration of innovations post-2000, reflecting renewed scientific interest in therapeutic ketosis.

**Figure 4 nutrients-17-02699-f004:**
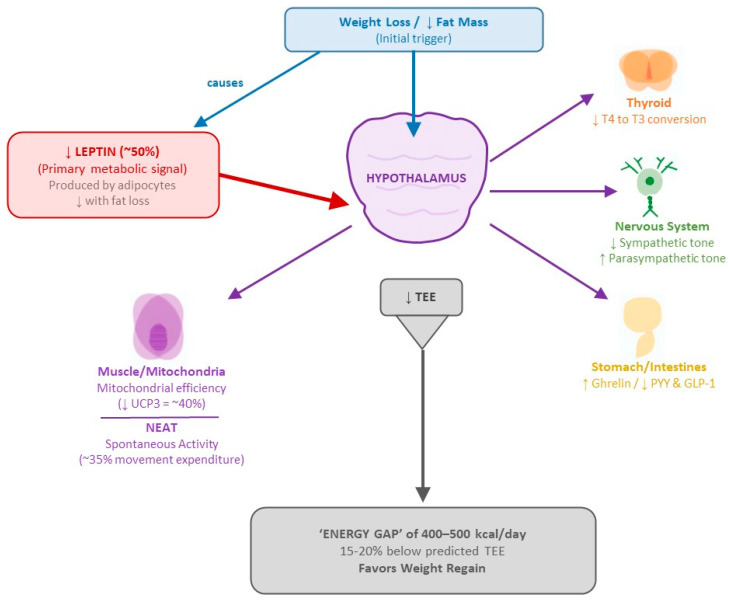
Metabolic adaptation cascade: coordinated physiological response to weight loss. The color and thickness of the arrows signify their role in the cascade. The thick blue arrow represents the initial physiological trigger. The extra-thick red arrow denotes the primary alarm signal (leptin drop). This triggers a complex neuroendocrine response, where the hypothalamus sends out coordinated downstream signals (purple arrows) to multiple effector systems. The final outcome—a suppressed TEE and the resulting ’Energy Gap’—is depicted in grey, representing the establishment of a new, persistent, and metabolically dampened state. Weight loss triggers a complex neuroendocrine cascade that actively defends body energy stores. The reduction in fat mass induces a disproportionate drop in circulating leptin (up to 50% with only 10% fat loss) [64], which acts as the primary signal to the hypothalamus. This relative hypoleptinemia orchestrates a coordinated systemic response, including: (1) Thyroid axis alterations: reduction in active T3 (from 112.7 ± 3.1 to 101.8 ± 2.6 ng/dL) without changes in TSH, along with a concomitant increase in reverse T3 [133,134]; (2) Autonomic changes: 20–40% fall in sympathetic nerve activity and increase in parasympathetic tone [136]; (3) Gut–brain axis dysregulation: persistent elevation of ghrelin and reduction of anorexigenic peptides (PYY, GLP-1), maintaining elevated hunger for at least 12 months [64,65]; (4) Muscular and mitochondrial adaptations: increased mitochondrial efficiency via a 40% reduction in uncoupling proteins (UCP3) [66,67], and a 35% decrease in non-exercise activity thermogenesis (NEAT) [68]. The integration of these mechanisms results in a 15–20% suppression of total energy expenditure (TEE) beyond what would be expected from changes in body composition [62], creating a persistent energy gap of 400–500 kcal/day. This adaptation may persist for years, as demonstrated in the 6-year follow-up of “The Biggest Loser” participants, where metabolic suppression remained at –499 ± 207 kcal/day [63], actively favoring weight regain and constituting the primary biological barrier to successful long-term weight loss maintenance.

**Figure 5 nutrients-17-02699-f005:**
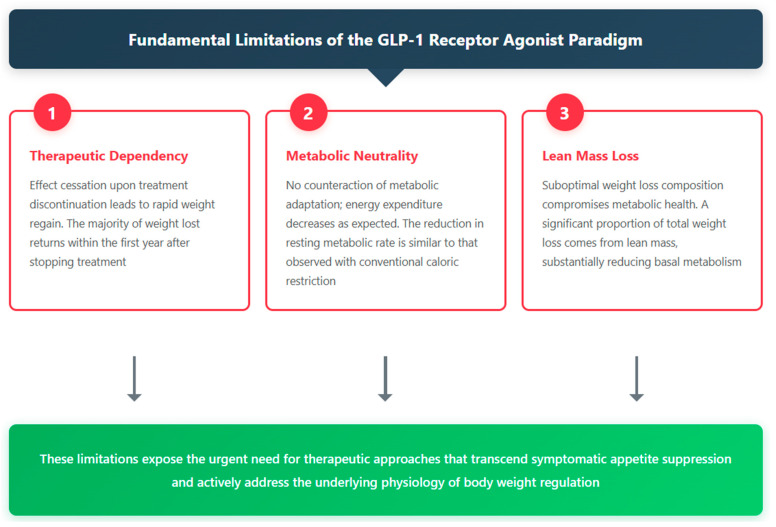
Intrinsic limitations of the pharmacological appetite suppression paradigm. The current GLP-1/GIP receptor agonist paradigm, despite its revolutionary clinical efficacy, presents three fundamental limitations that compromise long-term therapeutic success. The figure illustrates how these interconnected limitations—therapeutic dependency, metabolic neutrality, and lean mass loss—converge to highlight the urgent need for more comprehensive approaches. Therapeutic dependency manifests as rapid weight regain following treatment discontinuation, with the majority of lost weight returning within the first year [80]. Metabolic neutrality reflects the absence of any thermogenic advantage, as these agents do not counteract the adaptive reduction in energy expenditure that accompanies weight loss [81,153]. The loss of lean mass represents a particularly concerning aspect, as a significant proportion of total weight reduction comes from metabolically active tissue, further compromising basal metabolism [82]. These limitations collectively demonstrate that while GLP-1 receptor agonists effectively suppress appetite, they fail to address the underlying physiological mechanisms of weight regulation, underscoring the need for therapeutic paradigms that integrate appetite control with metabolic activation strategies.

**Figure 6 nutrients-17-02699-f006:**
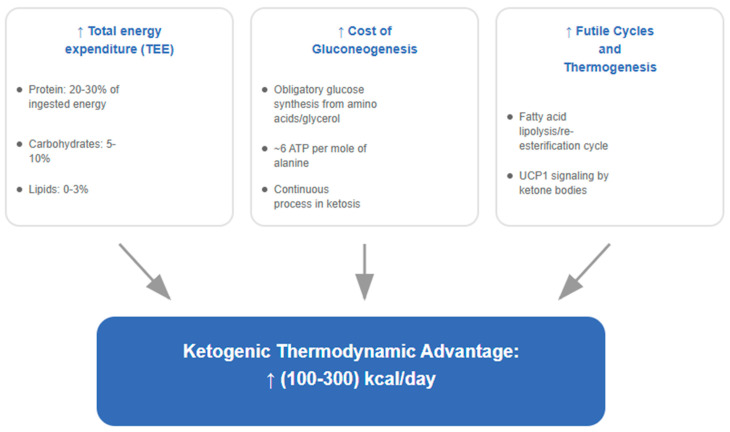
Mechanisms of ketogenic metabolic advantage. Three primary biochemical mechanisms contribute synergistically to the increased total energy expenditure (TEE) observed during nutritional ketosis, resulting in a metabolic advantage of 100–300 kcal/day. Top left: Enhanced thermic effect of food (TEF), with protein metabolism consuming 20–30% of ingested energy compared to 5–10% for carbohydrates and 0–3% for lipids [157]. Top center: Obligatory gluconeogenesis from amino acids and glycerol represents an ATP-consuming process that operates continuously during carbohydrate restriction, requiring approximately 6 ATP per mole of alanine converted to glucose, as cited in [158]. Top right: Futile cycles and ketone-induced thermogenesis, including the energetically costly fatty-acid lipolysis/re-esterification cycle [159] and experimental evidence that ketone bodies up-regulate UCP1 in murine brown adipose tissue [161]; confirmation in humans is pending. Bottom: The convergence of these mechanisms produces a measurable metabolic advantage that may partially offset the metabolic adaptation following weight loss. This thermodynamic inefficiency—reflecting the principles established by Fine and Feinman—may partially offset metabolic adaptation and provides a mechanistic basis whereby ketogenic diets could facilitate superior weight loss outcomes compared to higher-carbohydrate alternatives [83,84]. For clarity, evidence supporting each mechanistic pillar ranges from robust human data for TEF to predominantly animal data for UCP1 induction, a distinction that should be kept in mind when extrapolating to clinical practice. Evidence note: strongest in humans for protein-induced TEF; moderate for gluconeogenesis cost; limited for UCP1-mediated thermogenesis.

**Figure 7 nutrients-17-02699-f007:**
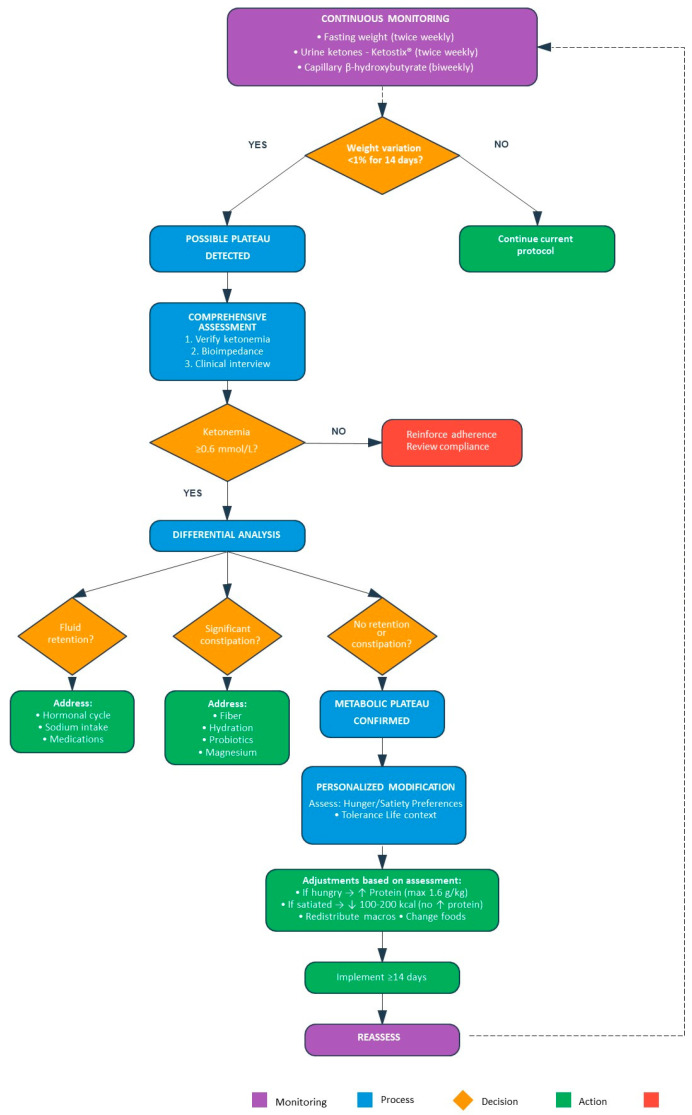
Dynamic adjustment system algorithm of the adaptive Ketogenic–Mediterranean Protocol (AKMP). Flowchart depicting the continuous monitoring and personalized adjustment cycle designed to detect and manage weight loss plateaus. The system differentiates between fluid retention, constipation, and true metabolic plateaus through comprehensive assessment, implementing individualized modifications based on hunger/satiety status. Adjustments include protein increases for hunger or caloric reductions for adequate satiety, maintained for ≥14 days before reassessment. The dashed line indicates the continuous monitoring cycle.

**Table 1 nutrients-17-02699-t001:** Comparative characteristics of the main ketogenic diet variants.

Variant and Acronym	Year/Key Author	Typical Macros (% C/P/F)	Main Application	Key Advantage (Mechanism/Benefit)	Main Limitation (Challenge)	Documented Results	Reference(s)
Classic Ketogenic Diet (CKD)	1921/Wilder	~4/~8/~90	Refractory Epilepsy	Maximum ketosis induction through strict 4:1 ratio	Highly restrictive, low adherence, requires strict medical supervision	>50% seizure reduction in drug-resistant epilepsy	[90]
Modified Ratios	Clinical evolution/1920s–1930s	~10/~10–15/~75–80	Pediatric Epilepsy	Greater nutritional flexibility while maintaining therapeutic efficacy (3:1, 2:1 ratios)	Still requires precise calculation and close monitoring	Efficacy comparable to 4:1 with better tolerance	[91]
MCT Diet (MCT)	1971/Huttenlocher	~15–20/~10/~70	Pediatric Epilepsy	Greater flexibility (more CHO/PRO) thanks to direct MCT metabolism	MCT oil cost, GI effects (diarrhea, nausea)	Similar efficacy to CKD with greater dietary variety	[92,93,94,95]
Original Atkins	1972/Atkins	Variable (20→100 g CHO)	Weight Loss	No caloric restriction, gradual CHO increase allows adaptation	Designed for weight loss, not for therapeutic ketosis	Rapid initial loss, variable long-term maintenance	[59]
Modified Atkins (MAD)	2003/Kossoff	~5/Ad libitum/Ad libitum	Epilepsy (Adolescents/Adults)	Easy implementation (no weighing/measuring, no caloric restriction), high adherence	Less control over body composition and macro quality	40–50% achieve >50% seizure reduction	[91,96]
Low Glycemic Index (LGIT)	2005/Pfeifer & Thiele	~10–20/~25/~60	Epilepsy	Allows more carbohydrates (40–60 g) if low GI, improving variety	Requires GI knowledge; less deep ketosis	>90% seizure reduction in 50% of patients	[97,106]
Very Low Calorie Ketogenic (VLCKD)	2000s/Castellana	~5/~20/~75 (<800 kcal)	Severe obesity	Very rapid weight loss with significant cardiometabolic improvements	Not sustainable long-term; requires strict medical supervision	↓ 15.6 kg, ↓ HbA1c 0.7%, ↓ TG 60 mg/dL, ↓ SBP 8 mmHg	[98]
Targeted (TKD)	1990s/Various	Standard + 15–30 g CHO peri-exercise	Endurance Athletes	Improves high-intensity performance without compromising keto-adaptation	Requires precise periodization; risk of exiting ketosis	↑ 2.8% time trial performance	[99]
Cyclical (cKD)	1990s/Various	5–6 d keto/1–2 d 60–70% CHO	Strength/Bodybuilding	Weekly muscle glycogen replenishment; maintains lean mass during deficit	Planning complexity; repeated keto/glycolysis adaptation	Preserves strength and muscle mass, ↓ body fat	[100]
High Protein (HPKD)	2000s/Westman	~5/~35/~60	Weight Loss/Type 2 Diabetes	Maximizes satiety and thermic effect; lean mass preservation; glycemic control	Risk of excessive gluconeogenesis if uncontrolled	Significantly greater weight loss vs. control diets (11.1 vs. 6.9 kg at 24 weeks); 95% ↓ DM2 medication	[101,102]
Eco-Atkins (Plant-based)	2009/Jenkins	~31/~26/~43	Dyslipidemia/Vegans	Enables ketosis in vegetarians/vegans; superior cardioprotective profile	Difficulty obtaining complete protein; dependence on processed foods	↓ CRP 28.2%, ↓ LDL 8.1%, ↓ 10.7 kg in 6 months	[103,104]
Paleolithic (PKD)	2010s/Clemens	~5–10/~20–25/~70–80	Autoimmunity/Clinical cases	Eliminates inflammatory foods; emphasis on organs and grass-fed meat	Very socially restrictive; evidence limited to cases	Anecdotal improvements in autoimmune conditions	[105]
Spanish Ketogenic Mediterranean (SKMD)	2008/Pérez-Guisado	~5/~20/~75	Obesity/CV Health	Combines keto + Mediterranean benefits; virgin olive oil as primary fat	Cost of quality Mediterranean ingredients	↓ 14.47 ± 6.88 kg (12 wk), ↓ TG 47.9%, ↑ HDL 10%	[69]
KEMEPHY	2011/Paoli	Variable (<50 g CHO)	Corporate weight loss	Prepared meals + phytoextracts; maximum convenience	Dependence on specific commercial products	92% adherence, ↓ 10 cm waist (6 wk)	[70]
Biphasic Ketogenic Mediterranean Diet	2013/Paoli	Alternates keto/Mediterranean	Long-term maintenance	Combines initial keto loss with Mediterranean sustainability	Not adaptive to individual response; fixed transitions	↓ 16.9 ± 7.1 kg/year, 88.2% adherence	[59]

Abbreviations: C/P/F: Carbohydrates/Protein/Fat; CHO: carbohydrates; CV: cardiovascular; DM2: Type 2 Diabetes Mellitus; GI: (in column 6), glycemic index (in diet names); MCT: medium-chain triglycerides; SBP: systolic blood pressure; CRP: C-reactive protein; TG: triglycerides. Note: Modified ratios (3:1, 2:1) were gradually developed by multiple medical centers as practical adaptations of the original 4:1 ratio to improve tolerance in specific populations, particularly pediatric patients and individuals with obesity.

**Table 2 nutrients-17-02699-t002:** Safety profile of well-formulated ketogenic diet: evidence vs. myth.

Area of Concern	Prevalent Myth	Scientific Evidence	Protective Mechanism	References
**Ketoacidosis**	“Ketosis is dangerous and can lead to ketoacidosis”	Nutritional ketosis (0.5–5 mmol/L) is physiological and safe. DKA (≥3 mmol/L) only occurs with absolute insulin deficiency	Intact insulin feedback prevents uncontrolled ketogenesis	[107,108,109]
**Cardiovascular Health**	“High fat intake increases cardiovascular risk”	Triglyceride reduction (−51%), HDL-C increase (+13%), blood pressure reduction, shift from sdLDL to lbLDL	Improved insulin sensitivity (−55%), reduced systemic inflammation	[110,111,114]
**Renal Function**	“KD damages kidneys due to high protein”	No deterioration in renal function; 27.7% normalization of GFR in mild CKD	KD is adequate, not high in protein; improves glycemic control and BP	[112]
**Muscle Mass**	“Ketosis causes muscle loss”	Complete preservation of strength in elite athletes; maintenance of explosive performance	Ketones reduce protein gluconeogenesis; increased growth hormone	[107,113]
**Bone Density**	“Ketosis causes bone deterioration”	No evidence of bone loss with well-formulated KD	Alkalinizing vegetables; reduced inflammation	[107]

Abbreviations: DKA, diabetic ketoacidosis; sdLDL, small dense LDL; lbLDL, large buoyant LDL; GFR, glomerular filtration rate; CKD, chronic kidney disease; KD, ketogenic diet; BP, blood pressure.

**Table 3 nutrients-17-02699-t003:** Contraindications and clinical precautions for the ketogenic diet.

Category	Specific Conditions	Clinical Considerations	References
Absolute Contraindications	• CPT I/II deficiency	Inability to metabolize fatty acids	[115,116,117]
	• Carnitine–acylcarnitine translocase deficiency	Mitochondrial transport impairment	[115,116]
	• Mitochondrial β-oxidation defects	Toxic metabolite accumulation	[115,117]
	• Acute intermittent porphyria	Risk of porphyric crisis	[115,120,121]
	• Pyruvate carboxylase deficiency	Gluconeogenesis impairment	[115,118,119]
	• Acute pancreatitis	Exacerbation by fat content	[115]
Special Precautions	• Type 1 Diabetes Mellitus	Mandatory basal/bolus insulin adjustment	[124,125,126]
	• SGLT-2i	Risk of euglycemic ketoacidosis	[122,123]
	• Antihypertensives	Possible need for dose reduction	[107]
	• Oral hypoglycemic agents	Adjustment according to blood glucose	[107,126]
Conditions Requiring Modification	• Pregnancy and lactation	Special nutritional requirements	[107]
	• CKD stage ≥ 4	Strict renal function monitoring	[112,115]
	• Hepatic insufficiency	Ketone metabolism impairment	[115]
	• Children < 2 years	Only under refractory epilepsy protocol	[115]
Recommended Monitoring	• Weeks 1–2	Daily capillary ketones	[115]
	• Monthly	Complete metabolic panel	[115]
	• Quarterly	Renal function, liver function, lipid profile	[112,115]

**Table 4 nutrients-17-02699-t004:** Pragmatic, safety-first medication-adjustment checklist during AKMP (for clinicians).

Drug/Class or Condition	Initial Action at AKMP Start	Ongoing Monitoring	Notes/Contingencies
Type 2 diabetes—SGLT-2 inhibitors	Discontinue prior to ketogenic induction to reduce euglycemic DKA risk; consider resumption in non-ketogenic phases if clinically indicated.	SMBG/CGM during Weeks 1–2; assess ketones if symptomatic.	Prefer alternatives during induction; re-evaluate if persistent hyperglycemia.
Type 2 diabetes—Insulin	Reduce basal by ~20–30%; withhold/reduce prandial insulin initially.	SMBG/CGM to titrate and avoid hypoglycemia.	Reinstate/titrate prandial doses as carb intake increases in later phases.
Type 2 diabetes—Sulfonylureas	Reduce dose by ~50% or discontinue.	SMBG; weekly review during Weeks 1–4.	Prioritize de-escalation if hypoglycemia or rapid weight loss.
Metformin/GLP-1 RA	Continue unless intolerance.	GI tolerance; weight trajectory.	Consider GLP-1 RA as adjunct in severe hyperphagia; reassess need over time.
Type 1 diabetes	Ketogenic phases only under specialist supervision; proactive basal/bolus adjustments.	Strict ketone-glucose monitoring and sick-day rules.	Abort ketogenic phase if persistent ketosis with hyperglycemia or intercurrent illness.
Hypertension/Diuretics	Expect early natriuresis; reassess thiazides/loops and ACEi/ARB doses.	Seated BP daily during Weeks 1–2; electrolytes if symptoms.	Adjust fluids/electrolytes per clinical status.
Lipid safety (atherogenic risk)	Baseline apoB (or LDL-P), LDL-C, non-HDL-C.	Repeat at 6–12 weeks.	If apoB rises substantially, shift fats toward MUFA/PUFA, add low-carb fiber/plant sterols; consider pharmacotherapy per guidelines.

Abbreviations: SMBG = self-monitoring of blood glucose; CGM = continuous glucose monitoring; apoB = apolipoprotein B; LDL-P = LDL particle number; MUFA = monounsaturated fatty acids; PUFA = polyunsaturated fatty acids. This checklist is pragmatic rather than prescriptive; decisions must be individualized and aligned with local standards.

**Table 5 nutrients-17-02699-t005:** Efficacy of GLP-1/GIP agonists in pivotal clinical trials.

Drug	Clinical Trial	n	Duration (weeks)	Weight Loss (%)	Weight Loss (kg)	Reference
Semaglutide	STEP 1	1961	68	14.9%	−15.3 kg	Wilding et al., 2021 [78]
Semaglutide	STEP 4	803	68	17.4%	−18.2 kg	Rubino et al., 2021 [148]
Tirzepatide	SURMOUNT-1	2539	72	20.9% (15 mg)	−22.2 kg	Jastreboff et al., 2022 [79]

**Table 6 nutrients-17-02699-t006:** Systematic analysis of Mediterranean Ketogenic Diet interventions: efficacy, adherence, and unaddressed metabolic adaptation.

Study	N	Duration	Protocol/Macros	Weight Loss	Adherence/Completion (%)	Adverse Events	Addresses Metabolic Adaptation?
Pérez-Guisado 2008 [69]	40	12 wk	SKMD: <30 g CHO, EVOO ≥ 30 mL/d	−14.1 kg (−13.0%)	77.5%	Mild headache, constipation	No
Paoli 2011 [70]	106	6 wk	KEMEPHY: <20 g CHO + phytoextracts	−7.0 kg	93.4%	Minimal	No
Paoli 2013 [59]	89	12 months	Biphasic: MKD/Med alternating	−16.1 kg (−16.0%)	88.8%	Transient initial phase	No
Cincione 2022 [71]	80	30 days	VLCKD-Med vs. VLCD-Med in T2DM	−7.2 kg	100%	Not significant	No
Gardner 2022 [72]	40	12 wk x2	Keto vs. Med-Plus crossover	−7–8% weight	90.0%	LDL elevation with keto	No
Sheffler 2023 [60]	58	6 wk	MKD + motivational support	NR	79–82%	None reported	No
Nagpal 2019 [75]	17	6 wk	Modified MKD in MCI	−3.1 kg	88.2%	Minimal	No

Abbreviations: SKMD = Spanish Ketogenic Mediterranean Diet; KEMEPHY = Ketogenic Mediterranean with Phytoextracts; MKD = Mediterranean Ketogenic Diet; VLCKD = Very Low-Calorie Ketogenic Diet; VLCD = Very Low-Calorie Diet; Med = Mediterranean; T2DM = Type 2 Diabetes Mellitus; MCI = Mild Cognitive Impairment; CHO = Carbohydrates; EVOO = Extra Virgin Olive Oil; NR = Not Reported.

**Table 7 nutrients-17-02699-t007:** Comparative matrix of weight management protocol design characteristics.

Evaluation Domain	Standard Hypocaloric Diet	GLP-1 Agonists	Standard KD	AKMP (Proposed Protocol)
Weight Quality (Lean Mass)	Lean mass loss of ~25% of total weight. No specific protective mechanisms [185].	Induces substantial weight loss composed of both fat mass and significant lean mass reduction, without active muscle protection mechanisms [82,186].	VLCKD preserves lean mass better than conventional hypocaloric diets [187,188,189].	Designed to preserve lean mass through protein adequacy (1.6 g/kg) and the anticatabolic effect of ketones.
Metabolic Adaptation (Expenditure)	Does not address TEE decline (see Section 4).	Potent appetite suppression, with reduced hunger and food preoccupation [190], without increased energy expenditure; in fact, REE decreases in line with expected metabolic adaptation [81].	Leverages metabolic advantage (150–300 kcal/day) but without dynamic adjustments [83,84].	Designed to actively counteract through: (1) Metabolic advantage, and (2) Dynamic adjustment system.
Sustainability and Adherence	Average long-term adherence of 60.5%. Improves with supervision (RR 1.65) [136].	Requires chronic treatment. Recovery of ~2/3 of weight lost upon discontinuation [80].	High adjusted adherence (93.4%) demonstrated in a 6-week supervised study [70]. Limited long-term sustainability.	Designed to maximize adherence through: (1) Mediterranean pattern, (2) Personalization, and (3) Intensive support.
Patient Autonomy	Moderate. Initial success predicts maintenance [191].	Minimal. Effect depends on continuous drug presence [80].	Moderate. Requires macro self-management.	Maximum. Central goal of empowering for self-management through Transition Phase and education.

Note: AKMP characteristics represent theoretical design objectives pending validation through controlled clinical trials.

## Data Availability

No new data were created or analyzed in this study. Data sharing is not applicable to this article.

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
