# Peer review of "Beyond GLP-1 Agonists: An Adaptive Ketogenic–Mediterranean Protocol to Counter Metabolic Adaptation in Obesity Management"

_nutrients, 2025, doi:10.3390/nu17162699_

Round 1

Reviewer 1 Report

Comments and Suggestions for Authors

This manuscript presents a comprehensive and well-structured review of the limitations of current obesity management strategies, particularly focusing on metabolic adaptation and adherence challenges. The authors propose an Adaptive Ketogenic-Mediterranean Protocol (AKMP) as a novel dietary intervention. The topic is timely and relevant, and the manuscript is rich in references and scientific rationale.

Strengths:  

- The manuscript is thorough and well-researched, with extensive citations from recent and high-impact studies.  

- The integration of ketogenic and Mediterranean dietary principles is innovative and well-justified.  

- The discussion on metabolic adaptation is detailed and highlights a critical gap in current obesity interventions.  

- The proposed AKMP is clearly described, with logical phases and mechanisms.

Weaknesses and Suggestions for Improvement:  

1. Empirical Validation:  

   The AKMP is presented as a theoretical construct. While the rationale is strong, the lack of clinical trial data limits its immediate applicability. The authors should emphasize this limitation more clearly and propose specific study designs for future validation.

2. Clarity and Conciseness:  

   Some sections, particularly the mechanistic explanations, are dense and may benefit from simplification for broader readership. Consider summarizing key points in tables or diagrams where appropriate.

3. Terminology Consistency:  

   Ensure consistent use of acronyms and terms throughout the manuscript (e.g., KMD, AKMP, VLCKD). A glossary or abbreviation list could improve readability.

4. Figures and Tables:  

   The manuscript includes several figures and tables, but some are overly complex. Simplifying visual elements and ensuring they are self-explanatory would enhance comprehension.

5. Ethical Considerations:  

   Although the manuscript is a review, the authors propose a protocol involving biomarker monitoring and dietary manipulation. A brief discussion on ethical considerations and patient safety would be beneficial.

Author Response

We sincerely thank Reviewer 1 for the thoughtful and constructive evaluation of our manuscript. We are grateful for the recognition of the manuscript’s scope, rationale, and the conceptual novelty of the Adaptive Ketogenic–Mediterranean Protocol (AKMP). We address below the items that prompted concrete revisions.

1) Empirical validation

We fully agree that the AKMP is, at present, a theoretical construct that requires prospective validation. We have made the following clarifications and additions:

  • In the Abstract, we now explicitly state that the AKMP requires confirmation in prospective, controlled studies and that we outline a pragmatic 24‑week pilot design (see Section 8.3.2).
  • Section 8.3.2 details a four-arm randomized pilot trial framework (AKMP→incretin, incretin→AKMP, tirzepatide alone, and usual care), including feasibility endpoints (adherence/retention), mechanistic assessments (REE by indirect calorimetry, DXA fat-free mass), safety monitoring, and handling of weight-loss plateaus under the AKMP’s dynamic algorithm.

2) Clarity and conciseness

According to your comment, authors have been the manuscript to improve the clarity and conciseness

3) Terminology consistency

We revised terminology to ensure consistent usage of acronyms across the manuscript and added a comprehensive Abbreviations section. In particular:

  • We now refer to the classical Ketogenic Diet as cKD (to avoid ambiguity with CKD, chronic kidney disease).
  • We consistently use KMD (Ketogenic–Mediterranean Diet) for the family of Mediterranean-ketogenic protocols and MMKD (modified Mediterranean–ketogenic diet) for the specific protocol employed by Nagpal et al. We also harmonized the label used for the Sheffler trial to “Mediterranean ketogenic dietary program + motivational support.”
  • These changes improve readability without altering the scientific content.

4) Supplementary Table S1 (29 KMD studies)

  • We apologize for any confusion. The complete list of the 29 KMD studies (study-level characteristics and outcomes) is now provided in Supplementary Table S1 and uploaded as a separate Supplementary file. We reinforced cross-references in Section 2.4 and Section 7.1, and we list all Supplementary materials in the final “Supplementary Materials” block to ensure visibility during review.
  • We thank the Reviewer again for the valuable feedback, which has improved the clarity, transparency, and terminological precision of our work.

5) Ethical considerations and patient safety

We have added a dedicated subsection (Section 7.6) outlining ethical safeguards for clinical implementation (informed consent, professional supervision, contraindications/precautions, and adherence to Good Clinical Practice for any future trials), including explicit caution with SGLT2 inhibitors and special populations.

Reviewer 2 Report

Comments and Suggestions for Authors

Peer Review – Major Revision Recommended

This manuscript proposes the Adaptive Ketogenic–Mediterranean Protocol (AKMP) as a novel, biomarker-guided dietary approach intended to enhance adherence and mitigate metabolic adaptation following weight loss. The conceptual framework is clinically relevant, and the narrative review offers a clear rationale for combining ketogenic and Mediterranean principles. However, significant methodological and reporting limitations need to be addressed before the paper is suitable for publication.

1. Systematic Review Rigor
The review section does not mention PROSPERO or other protocol registration, which is expected for transparency and to minimize bias. There is also no formal risk-of-bias or study-quality assessment, leaving readers without a measure of the reliability of the included evidence. A table summarizing all 29 identified studies — including design, population, duration, outcomes, ketosis measurement method, and quality rating — is absent and should be added to improve clarity and allow readers to assess heterogeneity.

2. Terminology and Limitations
The manuscript uses outdated terminology (e.g., “NAFLD” instead of “MASLD”), which should be updated in line with current consensus. The limitations section does not fully acknowledge language restriction bias (English-only) and exclusion of grey literature, nor the heterogeneity in definitions of “Ketogenic–Mediterranean Diet.” These omissions need correction to improve transparency.

3. AKMP Description and Measurement Strategy
Several design elements require clarification and refinement:

  • Justify the 14-day plateau window — specify whether it is empirical or pragmatic, and consider sensitivity analysis in future testing (10 vs 14 vs 21 days).

  • Replace “Ketostix as high-frequency measure” with a hybrid approach: daily (or alternate-day) blood β-hydroxybutyrate during induction, then urine strips as adjunct, noting their limitations.

  • Distinguish between clinical-practice monitoring (pragmatic tools) and research-grade monitoring (gold standards such as DXA for body composition, DLW for TEE, indirect calorimetry for REE). If InBody BIA is used, validate against DXA in a pilot.

  • When monitoring safety, measure lipoprotein subfractions rather than only LDL-C.

4. Mechanistic Assumptions and Safety
The stated metabolic advantage of ketosis (200–300 kcal/day) should be presented as probable but context-dependent, with caveats regarding measurement methods, study durations, and population variability. Safety considerations for SGLT-2 inhibitors and type 1 diabetes are noted, but a more explicit, algorithmic medication-adjustment plan is recommended.

Conclusion
The AKMP is a promising concept, but methodological transparency, up-to-date terminology, complete limitation reporting, and more rigorous description of monitoring and mechanistic claims are essential. Addressing the above points will markedly improve the scientific robustness and clinical credibility of the manuscript.

Author Response

First and foremost, we would like to express our sincere gratitude for your time and for the detailed, constructive feedback on our manuscript. Your suggestions have been invaluable, and we believe they have significantly improved the scientific robustness, clarity, and clinical credibility of our paper.

We have addressed each of your points and have detailed the corresponding modifications below:

  1. Regarding the Rigor of the Review
  • We have clarified in the Methods section (paragraphs 188 and 238) why PROSPERO registration was not applicable, justifying the nature of our work as a "thesis-driven, argumentative narrative review" rather than a conventional systematic review.
  • Following this rationale, we have explicitly stated that a formal risk-of-bias assessment was not undertaken, and we have added a discussion of this choice as a limitation in Section 8.4 for greater transparency.
  • We have included the requested Supplementary Table S1, which now contains a detailed, study-level summary of the 29 identified KMD protocols, with information on design, population, duration, outcomes, and ketosis verification method, as you suggested.
  1. Regarding Terminology and Limitations
  • We have updated all outdated terminology, replacing "NAFLD" with the current consensus term, "MASLD," and defining this change upon its first mention in the text.
  • Furthermore, we have expanded Section 8.4 (Limitations) to explicitly and more thoroughly acknowledge the potential for language-restriction bias, the exclusion of grey literature, and the heterogeneity in the definitions of the "Ketogenic-Mediterranean Diet" across studies.
  1. Regarding the AKMP Description and Measurement Strategy

We have refined several elements of the AKMP design for greater clarity and rigor:

  • Justification of the 14-day plateau window: We have specified the pragmatic and evidence-based rationale for this timeframe in Section 7.4. We also incorporated your suggestion to consider a sensitivity analysis (10 vs. 14 vs. 21 days) in the future pilot trial proposal (Section 8.3.2).
  • Ketone measurement strategy: We have adopted the hybrid approach you recommended, clearly distinguishing in Section 7.4 between monitoring for clinical practice (pragmatic, low-burden tools) and for research settings (gold-standard assessments like DXA, DLW, and calorimetry).
  • Lipid safety: We have incorporated the recommendation to measure lipoprotein subfractions. We now specify the prioritization of apoB (or LDL−P) for monitoring atherogenic risk in both Section 3.2.2 and the new medication-adjustment checklist (Box 1).
  1. Regarding Mechanistic Assumptions and Safety
  • We have nuanced our description of the "metabolic advantage," presenting it throughout the text as a "context-dependent" effect, whose precise magnitude remains debated and varies by measurement method and study duration, as discussed in Section 6.1.
  • We have added an explicit, algorithmic medication-adjustment plan. The new "Box 1. Pragmatic, safety-first medication-adjustment checklist during AKMP (for clinicians)" provides clear, practical guidance for managing key medications (SGLT−2 inhibitors, insulin, antihypertensives, etc.) during the protocol's implementation.

Thank you once again for your thorough review.

Reviewer 3 Report

Comments and Suggestions for Authors In the present manuscript, Cayetano García-Gorrita and coworkers systematically reviewed Ketogenic Mediterranean Diet (KMD) protocols, analyzed why they fail to counteract metabolic adaptation and presented the Adaptive Ketogenic–Mediterranean Protocol (AKMP). The authors concluded that the AKMP integrates: i) a Mediterranean base to enhance adherence, ii) a ketogenic engine for thermodynamic advantage, and iii) a dynamic biomarker-based adjustment system. So far, it represents the first intervention explicitly designed to simultaneously address the behavioral and biological challenges of weight maintenance. Overall, I think that the manuscript is very intriguing, timely (within the scope of "Nutrients”), and of clinical impact on a current topic of interest. So far, in my humble opinion, I have some suggestions to improve the quality of the present manuscript. 1) It may be appropriate to register this systematic review in a public register (for example, Research registry, Open Science, JBI etc.) where the authors further certify the compliance of this systematic review with PRISMA guidelines. 2) In light of the clinical trials here included, please discuss, in the revised version of manuscript, the possible therapeutic application of soy isoflavones (i.e., Circulation 2020, 141, 1127-1137; Nutrients, 2022, 14, 1550), in combination with AMKP and a regular physical activity, that, particularly in post-menopausal women, could provide a further alternative strategy to address the behavioral and biological challenges of weight maintenance, and delay the progression of Noncommunicable diseases. 3) Tirzepatide is a once-weekly glucagon-like peptide 1 (GLP-1) and glucose-dependent-insulinotropic-polypeptide (GIP) receptor dual agonist approved for type 2 diabetes in several countries. Furthermore, Tirzepatide showed excellent results in terms of improvement of glucose control and weight loss, as indicated by preclinical studies and clinical trials, attributable to the synergistic effects of the concomitant dual agonism on GIP and GLP-1 receptors. Please discuss this crucial aspect considering the AKMP here proposed.

Author Response

We sincerely thank you for your positive evaluation and constructive suggestions. Below we address each point and indicate exactly where changes were implemented in the revised manuscript.

1) Registration / PRISMA compliance

Comment. You suggested registering the review in a public registry and certifying PRISMA compliance.

Response. We clarified the hybrid nature of our work (argumentative, thesis-driven narrative review anchored by a focused evidence-mapping component rather than a conventional PICO-based systematic review). Consequently, prospective registration (e.g., PROSPERO) is not applicable. To strengthen transparency, we (i) reframed the Abstract and Methods to say “focused evidence-mapping,” and (ii) added a PRISMA-style flow diagram for the mapping component (Figure 2).

Where updated. Abstract—Objectives & Methods; §2 Materials and Methods (second paragraph); subsection Reporting Framework and Transparency; Figure 2.

2) Soy isoflavones (post-menopausal women) in combination with AKMP and physical activity

Comment. You asked us to discuss potential therapeutic application of soy isoflavones together with AKMP and exercise.

Response. We integrated a concise, clinically framed paragraph highlighting potential synergy for post-menopausal women within a personalized AKMP framework. We cite the two sources you recommended:

  • Circulation (2020) — “Isoflavone Intake and the Risk of Coronary Heart Disease in US Men and Women” (Ma et al., 141:1127–1137).
  • Nutrients (2022) — “Mediterranean Diet and Soy Isoflavones for Integrated Management of the Menopausal Metabolic Syndrome” (Marini, 14:1550).

Where added. §8.3 Integration with Pharmacotherapy (final paragraphs on personalization for post-menopausal women). We also list these items in the References.

3) Tirzepatide (dual GIP/GLP-1 agonism) and synergy—implications for AKMP

Comment. You requested explicit discussion of the synergistic mechanism and how it relates to AKMP.

Response. We strengthened our incretin section to emphasize the complementary, possibly synergistic effects of concomitant GIP and GLP-1 receptor agonism, and we discuss the clinical signal from SURMOUNT-1 (“Tirzepatide Once Weekly for the Treatment of Obesity,” N Engl J Med, 2022) alongside mechanistic considerations. We then make explicit how AKMP can (i) preserve lean mass through adequate protein and the anti-catabolic milieu of ketosis and (ii) counter incretin “metabolic neutrality” by leveraging the ketogenic metabolic advantage during weight loss and maintenance.

Where updated. §5.1 GLP-1/GIP Agonists (mechanistic emphasis and clinical context); §8.3 (last paragraph); new §8.3.1 detailing two pragmatic sequences—AKMP-first with step-up to incretin therapy, and incretin-first with tapered carbohydrate reduction into AKMP.

Transparency note. The evidence-mapping of Ketogenic–Mediterranean protocols now culminates in Supplementary Table S1 (n = 29 studies) with study-level details, improving reproducibility and addressing traceability of included protocols.

Where noted. 2.4; 3.1; Supplementary Table S1.

We hope these revisions fully address your suggestions and improve clinical interpretability while remaining faithful to the thesis-driven purpose of the article.

Reviewer 4 Report

Comments and Suggestions for Authors

Interesting idea of this study, my recommendations are the following:
Abstract I recommend that the conclusions mention the differences between the two aspects targeted, also taking into account GLP-1 receptor agonists.
Section 1.3.2. lines 99-125 mention the bibliographic index in almost all the ideas presented, I recommend focusing on the aspects you want to highlight. And introducing other bibliographic indexes to support the statements.
Lines 143-144 I recommend that the acronyms be mentioned in parentheses and to present descriptively what they represent.
Lines 153-159 are conclusions, I recommend moving them to the specific section without duplicating information.
Fig 3 is not clear, I recommend enlarging it.
Supplementary Table S1 is not attached, I recommend clarifications.
Fig 7 is not readable, it is unclear, I recommend enlarging it.
8. General Discussion, Limitations, and Future Directions- I recommend presenting only the aspects not previously mentioned, focused.
In conclusion, your study is very well organized, logically presented, up-to-date but certain details are too elaborately presented, what I recommend to the authors is to revise it because it is very long.
I recommend mentioning the bibliography according to the editing rules.

Author Response

Thank you for your thoughtful and constructive feedback, which has helped us significantly improve the manuscript's focus and clarity. We have carefully considered your comments and have implemented the following revisions:

  • Abstract: The abstract’s conclusion now explicitly contrasts the standard pharmacological approach (i.e., GLP-1 receptor agonists) with our proposed dietary–metabolic architecture (the AKMP), highlighting its unique integration of a Mediterranean foundation, a ketogenic engine, and biomarker-guided adjustments.
  • Mechanistic Focus & Citations (Section 1.3.2): We have sharpened the mechanistic argument in this section and diversified our references. We now incorporate evidence on reward-pathway activation after high-glycemic meals, reduced striatal D2 receptor availability in obesity, and the distinct roles of food processing, fat, and glycemic load in addictive-like eating patterns. A brief note has also been added to clarify the clinical overlap between binge-eating disorder (BED) and food addiction.
  • Acronyms & Terminology: To ensure clarity and adherence to current standards, we now use the updated term “metabolic dysfunction–associated steatotic liver disease (MASLD; formerly non-alcoholic fatty liver disease, NAFLD)” at first mention. We have also performed a thorough check to ensure all acronyms are correctly defined upon their first use throughout the manuscript.
  • Figure Readability (Figures 3 & 7): Both figures have been regenerated at a higher resolution and enlarged to ensure they are clear and legible.
  • Supplementary Table S1: We have re-attached Supplementary Table S1 as requested.
  • General Discussion (Section 8): This section has been restructured to provide a more focused narrative. It now opens by framing the obesity epidemic as a ‘perfect storm’ resulting from the interplay between the modern food environment, an obesogenic context, and individual biological susceptibility. The discussion now concentrates on novel aspects not detailed earlier in the manuscript, including a frank assessment of limitations, specific testable hypotheses contrasting our AKMP with GLP-1/GIP therapies, and the critical role of biomarker-guided personalization.
  • Genetic Evidence (GWAS): To strengthen the manuscript's evidence base, the statement on >900 BMI-associated loci is now anchored with citations to the large-scale GWAS meta-analyses by Locke et al. (2015) and Yengo et al. (2018).

We are grateful for your valuable input and are confident that these revisions have substantially improved the quality and impact of our manuscript.

Round 2

Reviewer 2 Report

Comments and Suggestions for Authors

The authors addressed all my comments. Manuscript can now be accepted to the best of my knowledge.

Best regards